# BENCHMARKING UNSUPERVISED OBJECT REPRESENTATIONS FOR VIDEO SEQUENCES

## ABSTRACT

Perceiving the world in terms of objects and tracking them through time is a crucial prerequisite for reasoning and scene understanding. Recently, several methods have been proposed for unsupervised learning of object-centric representations. However, since these models have been evaluated with respect to different downstream tasks, it remains unclear how they compare in terms of basic perceptual abilities such as detection, figure-ground segmentation and tracking of individual objects. To close this gap, we design a benchmark with three datasets of varying complexity and seven additional test sets which feature challenging tracking scenarios relevant for natural videos. Using this benchmark, we compare the perceptual abilities of four unsupervised object-centric learning approaches: ViMON, a video-extension of MONET, based on a recurrent spatial attention mechanism, OP3, which exploits clustering via spatial mixture models, as well as TBA and SCALOR, which use an explicit factorization via spatial transformers. Our results suggest that architectures with unconstrained latent representations and full-image object masks such as ViMON and OP3 are able to learn more powerful representations in terms of object detection, segmentation and tracking than the explicitly parameterized spatial transformer based architecture of TBA and SCALOR. We also observe that none of the methods are able to gracefully handle the most challenging tracking scenarios despite their synthetic nature, suggesting that our benchmark may provide fruitful guidance towards learning more robust object-centric video representations.

## 1 INTRODUCTION

Humans understand the world in terms of objects. Being able to decompose our environment into independent objects that can interact with each other is an important prerequisite for reasoning and scene understanding. Similarly, an artificial intelligence system would benefit from the ability to both extract objects and their interactions from video streams, and keep track of them over time.

Recently, there has been an increased interest in *unsupervised* learning of *object-centric representations*. The key insight of these methods is that the compositionality of visual scenes can be used to both discover (Eslami et al., 2016; Greff et al., 2019; Burgess et al., 2019) and track objects in videos (Greff et al., 2017; van Steenkiste et al., 2018; Veerapaneni et al., 2019) without supervision. However, it is currently not well understood how the learned visual representations of different models compare to each other quantitatively, since the models have been developed with different downstream tasks in mind and have not been evaluated using a common protocol. Hence, in this work, we propose a benchmark based on procedurally generated video sequences to test basic perceptual abilities of object-centric video models under various challenging tracking scenarios.

An unsupervised object-based video representation should *(1)* effectively identify objects as they enter a scene, *(2)* accurately segment objects, as well as *(3)* maintain a consistent representation for each individual object in a scene over time. These perceptual abilities can be evaluated quantitatively in the established *multi-object tracking* framework (Bernardin & Stiefelhagen, 2008; Milan et al., 2016). We propose to utilize this protocol for analyzing the strengths and weaknesses of different object-centric representation learning methods, independent of any specific downstream task, in order to uncover the different inductive biases hidden in their choice of architecture and loss formulation. We therefore compiled a benchmark consisting of three procedurally generated video datasets of varying levels of

visual complexity and two generalization tests. Using this benchmark, we quantitatively compared three classes of object-centric models, leading to the following insights:

- All of the models have shortcomings handling occlusion, albeit to different extents.
- OP3 (Veerapaneni et al., 2019) performs strongest in terms of quantitative metrics, but exhibits a surprisingly strong dependency on color to separate objects and accumulates false positives when fewer objects than slots are present.
- Spatial transformer models, TBA (He et al., 2019) and SCALOR (Jiang et al., 2020), train most efficiently and feature explicit depth reasoning in combination with amodal masks, but are nevertheless outperformed by the simpler model, VIMON, lacking a depth or interaction model, suggesting that the proposed mechanisms may not yet work as intended.

We will make our code, data, as well as a public leaderboard of results available.

## 2 RELATED WORK

Several recent lines of work propose to learn object-centric representations from visual inputs for static and dynamic scenes without explicit supervision. Though their results are promising, methods are currently restricted to handling synthetic datasets and as of yet are unable to scale to complex natural scenes. Furthermore, a systematic quantitative comparison of methods is lacking.

Selecting and processing parts of an image via *spatial attention* has been one prominent approach for this task (Mnih et al., 2014; Eslami et al., 2016; Kosiorek et al., 2018; Burgess et al., 2019; Yuan et al., 2019; Crawford & Pineau, 2019; Locatello et al., 2020). As an alternative, *spatial mixture* models decompose scenes by performing image-space clustering of pixels that belong to individual objects (Greff et al., 2016; 2017; 2019; van Steenkiste et al., 2018). While some approaches aim at learning a suitable representation for downstream tasks (Watters et al., 2019a; Veerapaneni et al., 2019), others target scene generation (Engelcke et al., 2019; von Kügelgen et al., 2020). We analyze three classes of models for processing videos, covering three models based on spatial attention and one based on spatial mixture modeling.

**Spatial attention models with unconstrained latent representations** use per-object variational autoencoders, as introduced by Burgess et al. (2019). von Kügelgen et al. (2020) adapts this approach for scene generation. So far, such methods have been designed for static images, but not for videos. We therefore extend MONET (Burgess et al., 2019) to be able to accumulate evidence over time for tracking, enabling us to include this class of approaches in our evaluation. Recent concurrent work on AlignNet (Creswell et al., 2020) applies MONET frame-by-frame and tracks objects by subsequently ordering the extracted objects consistently.

**Spatial attention models with factored latents** use an explicit factorization of the latent representation into properties such as position, scale and appearance (Eslami et al., 2016; Crawford & Pineau, 2019). These methods use spatial transformer networks (Jaderberg et al., 2015) to render per-object reconstructions from the factored latents (Kosiorek et al., 2018; He et al., 2019; Jiang et al., 2020). SQAIR (Kosiorek et al., 2018) does not perform segmentation, identifying objects only at the bounding-box level. We select Tracking-by-Animation (TBA) (He et al., 2019) and SCALOR (Jiang et al., 2020) for analyzing spatial transformer methods in our experiments, which explicitly disentangle object shape and appearance, providing access to object masks.

**Spatial mixture models** cluster pixels using a deep neural network trained with expectation maximization (Greff et al., 2017; van Steenkiste et al., 2018). IODINE (Greff et al., 2019) extends these methods with an iterative amortised variational inference procedure (Marino et al., 2018), improving segmentation quality. SPACE (Lin et al., 2020) combines mixture models with spatial attention to improve scalability. To work with video sequences, OP3 (Veerapaneni et al., 2019) extends IODINE by modeling individual objects' dynamics as well as pairwise interactions. We therefore include OP3 in our analysis as a representative spatial mixture model.

## 3 OBJECT-CENTRIC REPRESENTATION BENCHMARK

To compare the different object-centric representation learning models on their basic perceptual abilities, we use the well-established multi-object tracking (MOT) protocol (Bernardin & Stiefelhagen,

Table 1: Summary of datasets and example video sequences. See Appendix B for details.

| | Objects | | | | | | Background | |
| Dataset | Shape | Motion | Count Over Sequence | Size Variation | Orientation | Color | Motion | Color |
|---|---|---|---|---|---|---|---|---|
| SpMOT | 4 Templates (2D) | Linear | Varies (0-3) | Minimal | Fixed | 6 Colors | None | Black |
| VOR | 2 Templates (3D) | Static | Varies (0-4) | Moderate | Random | 6 Colors | Moving Camera | Random |
| VMDS | 3 Templates (2D) | Non-Linear | Fixed (1-4) | Moderate | Random | $256^3$ Colors | None | Random |

2008). In this section, we describe the datasets and metrics considered in our benchmark, followed by a brief description of the models evaluated.

## 3.1 DATASETS

Current object-centric models are not capable of modeling complex natural scenes (Burgess et al., 2019; Greff et al., 2019; Lin et al., 2020). Hence, we focus on synthetic datasets that resemble those which state-of-the-art models were designed for. Specifically, we evaluate on three synthetic datasets[1] (see Table 1), which cover multiple levels of visual and motion complexity. Synthetic stimuli enable us to precisely generate challenging scenarios in a controllable manner in order to disentangle sources of difficulty and glean insights on what models specifically struggle with. We design different scenarios that test complexities that would occur in natural videos such as partial or complete occlusion as well as similar object appearances.

**Sprites-MOT** (**SpMOT**, Table 1 left), as proposed by He et al. (2019), features simple 2D sprites moving linearly on a black background with objects moving in and out of frame during the sequence. **Video-Multi-dSprites** (**VMDS**, Table 1 right) is a video dataset we generated based on a colored, multi-object version of the dSprites dataset (Matthey et al., 2017). Each video contains one to four sprites that move non-linearly and independently of each other with the possibility of partial or full occlusion. Besides the i.i.d. sampled training, validation and test sets of VMDS, we generate seven additional challenge sets that we use to study specific test situations we observed to be challenging, such as guaranteed occlusion, specific object properties, or out-of-distribution appearance variations. **Video Objects Room** (**VOR**, Table 1 middle) is a video dataset we generated based on the static Objects Room dataset (Greff et al., 2019), which features static objects in a 3D room with a moving camera. For full details on the datasets and their generation, see Appendix B.

## 3.2 METRICS

Our evaluation protocol follows the multi-object tracking (MOT) challenge, a standard and widely-used benchmark for supervised object tracking (Milan et al., 2016). The MOT challenge uses the CLEAR MOT metrics (Bernardin & Stiefelhagen, 2008), which quantitatively evaluate different performance aspects of object detection, tracking and segmentation. To compute these metrics, predictions have to be matched to ground truth. Unlike Bernardin & Stiefelhagen (2008) and Milan et al. (2016), we use binary segmentation masks for this matching instead of bounding boxes, which helps us better understand the models' segmentation capabilities. We consider an intersection over union (IoU) greater than 0.5 as a match (Voigtlaender et al., 2019). The error metrics used are the fraction of **Misses (Miss)**, **ID switches (ID S.)** and **False Positives (FPs)** relative to the number of ground truth masks. In addition, we report the **Mean Squared Error (MSE)** of the reconstructed image outputs summed over image channels and pixels.

To quantify the overall number of failures, we use the **MOT Accuracy (MOTA)**, which measures the fraction of all failure cases compared to the total number of objects present in all frames. A model with 100% MOTA perfectly tracks all objects without any misses, ID switches or false positives. To quantify the segmentation quality, we define **MOT Precision (MOTP)** as the average IoU of segmentation masks of all matches. A model with 100% MOTP perfectly segments all tracked objects, but does not necessarily track all objects. Further, to quantify detection and tracking performance

---

[1]Datasets are available at this https URL.

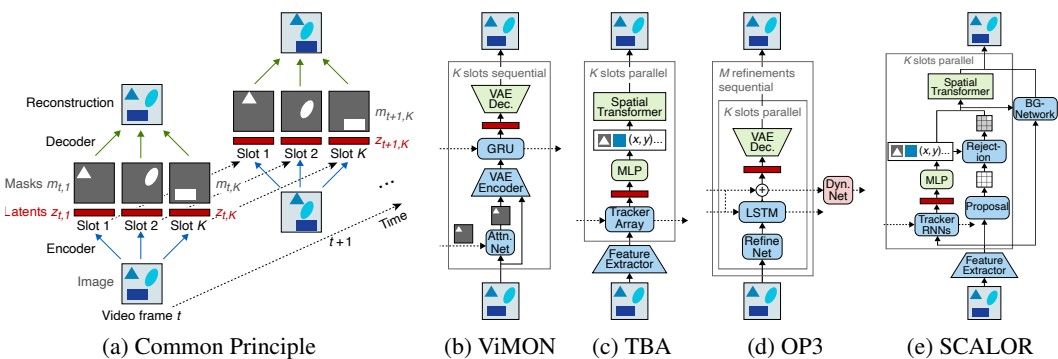

Figure 1: Common principles of all models: Decomposition of an image into a fixed number of slots, each of which contains an embedding $z_{t,k}$ and a mask $m_{t,k}$ of (ideally) a single object. Dotted lines: temporal connections. Solid lines: information flow within one frame.

independent of false positives, we measure the **Mostly Detected (MD)** and **Mostly Tracked (MT)** metrics, the fraction of ground truth objects that have been detected and tracked for at least 80% of their lifespan, respectively. If an ID switch occurs, an object is considered detected but not tracked. For full details regarding the matching process and the evaluation metrics, refer to Appendix A.

## 3.3 MODELS

We consider three classes of unsupervised object-centric representation learning models: *(1)* a spatial attention model with unconstrained latents, VIMON, which is our video extension of MONET (Burgess et al., 2019); *(2)* spatial transformer-based attention models, TBA (He et al., 2019) and SCALOR (Jiang et al., 2020); *(3)* a scene mixture model, OP3 (Veerapaneni et al., 2019). At a high-level, these methods share a common structure which is illustrated in Fig. 1a. They decompose an image into a fixed number of *slots* (Burgess et al., 2019), each of which contains an embedding $\mathbf{z}_{t,k}$ and a mask $\mathbf{m}_{t,k}$ of (ideally) a single object. These slots are then combined in a decoding step to reconstruct the image. Below, we briefly describe each method. Appendix C provides a detailed explanation in a unified mathematical framework.

**Video MONet (VIMON)** is our video extension of MONET (Burgess et al., 2019). MONET recurrently decomposes a static scene into slots, using an attention network to sequentially extract attention masks $\mathbf{m}_k \in [0,1]^{H \times W}$ of individual objects $k$. A Variational Autoencoder (VAE) (Kingma & Welling, 2014) encodes each slot into a latent representation $\mathbf{z}_k \in \mathbb{R}^L$ of the corresponding object. We use MONET as a simple frame-by-frame baseline for detection and segmentation that does not employ temporal information. VIMON accumulates evidence about the objects over time to maintain a consistent object-slot assignment throughout the video. This is achieved by *(1)* seeding the attention network the predicted mask $\widehat{\mathbf{m}}_{t,k} \in [0,1]^{H \times W}$ from the previous time step and *(2)* introducing a gated recurrent unit (GRU) (Cho et al., 2014), which aggregates information over time for each slot separately, enabling it to encode motion information. For full details on MONET and VIMON, as well as ablations to provide context for the design decisions, refer to Appendix C.1, C.2 and E.3.

**Tracking-by-Animation (TBA)** (He et al., 2019) is a spatial transformer-based attention model. Frames are encoded by a convolutional feature extractor $f$ before being passed to a recurrent block $g$ called Reprioritized Attentive Tracking (RAT). RAT re-weights slot input features based on their cosine similarity with the slots from the previous time step and outputs latent representations for all $K$ slots in parallel. Each slot latent is further decoded into a mid-level representation $\mathbf{y}_{t,k}$ consisting of pose and depth parameters, as well as object appearance and shape templates (see Fig. 1c). For rendering, a Spatial Transformer Network (STN) (Jaderberg et al., 2015) is used with an additional occlusion check based on the depth estimate. TBA is trained on frame reconstruction with an additional penalty for large object sizes to encourage compact bounding boxes. TBA can only process scenes with static backgrounds, as it preprocesses sequences using background subtraction (Bloisi & Iocchi, 2012). For full details on TBA, refer to Appendix C.3.

**Object-centric Perception, Prediction, and Planning (OP3)** (Veerapaneni et al., 2019) extends IODINE (Greff et al., 2019) to operate on videos. IODINE decomposes an image into objects and

Table 2: Analysis of SOTA object-centric representation learning models for MOT. Results shown as mean $\pm$ standard deviation of three runs with different random training seeds.

| Model | MOTA ↑ | MOTP ↑ | MD ↑ | MT ↑ | Match ↑ | Miss ↓ | ID S. ↓ | FPs ↓ | MSE ↓ |
|---|---|---|---|---|---|---|---|---|---|
| | | | | | **SpMOT** | | | | |
| MONET | 70.2 ± 0.8 | 89.6 ± 1.0 | 92.4 ± 0.6 | 50.4 ± 2.4 | 75.3 ± 1.3 | 4.4 ± 0.4 | 20.3 ± 1.6 | 5.1 ± 0.5 | 13.0 ± 2.0 |
| VIMON | 92.9 ± 0.2 | **91.8 ± 0.2** | 87.7 ± 0.8 | 87.2 ± 0.8 | 95.0 ± 0.2 | 4.8 ± 0.2 | **0.2 ± 0.0** | 2.1 ± 0.1 | 11.1 ± 0.6 |
| TBA | 79.7 ± 15.0 | 71.2 ± 0.3 | 83.4 ± 9.7 | 80.0 ± 13.6 | 87.8 ± 9.0 | 9.6 ± 6.0 | 2.6 ± 3.0 | 8.1 ± 6.0 | 11.9 ± 1.9 |
| OP3 | 89.1 ± 5.1 | 78.4 ± 2.4 | 92.4 ± 4.0 | 91.8 ± 3.8 | **95.9 ± 2.2** | 3.7 ± 2.2 | 0.4 ± 0.0 | 6.8 ± 2.9 | 13.3 ± 11.9 |
| SCALOR | **94.9 ± 0.5** | 80.2 ± 0.1 | **96.4 ± 0.1** | **93.2 ± 0.7** | 95.9 ± 0.4 | **2.4 ± 0.0** | 1.7 ± 0.4 | **1.0 ± 0.1** | **3.4 ± 0.1** |
| | | | | | **VOR** | | | | |
| MONET | 37.0 ± 6.8 | 81.7 ± 0.5 | 76.9 ± 2.2 | 37.3 ± 7.8 | 64.4 ± 5.0 | 15.8 ± 1.6 | 19.8 ± 3.5 | 27.4 ± 2.3 | 12.2 ± 1.4 |
| VIMON | **89.0 ± 0.0** | **89.5 ± 0.5** | **90.4 ± 0.5** | **90.0 ± 0.4** | **93.2 ± 0.4** | **6.5 ± 0.4** | 0.3 ± 0.0 | 4.2 ± 0.4 | 6.4 ± 0.6 |
| OP3 | 65.4 ± 0.6 | 89.0 ± 0.6 | 88.0 ± 0.6 | 85.4 ± 0.5 | 90.7 ± 0.3 | 8.2 ± 0.4 | 1.1 ± 0.2 | 25.3 ± 0.6 | **3.0 ± 0.1** |
| SCALOR | 74.6 ± 0.4 | 86.0 ± 0.2 | 76.0 ± 0.4 | 75.9 ± 0.4 | 77.9 ± 0.4 | 22.1 ± 0.4 | **0.0 ± 0.0** | **3.3 ± 0.2** | 6.4 ± 0.1 |
| | | | | | **VMDS** | | | | |
| MONET | 49.4 ± 3.6 | 78.6 ± 1.8 | 74.2 ± 1.7 | 35.7 ± 0.8 | 66.7 ± 0.7 | 13.6 ± 1.0 | 19.7 ± 0.6 | 17.2 ± 3.1 | 22.2 ± 2.2 |
| VIMON | 86.8 ± 0.3 | 86.8 ± 0.0 | 86.2 ± 0.3 | 85.0 ± 0.3 | 92.3 ± 0.2 | 7.0 ± 0.2 | 0.7 ± 0.0 | 5.5 ± 0.1 | 10.7 ± 0.1 |
| TBA | 54.5 ± 12.1 | 75.0 ± 0.9 | 62.9 ± 5.9 | 58.3 ± 6.1 | 75.9 ± 4.3 | 21.0 ± 4.2 | 3.2 ± 0.3 | 21.4 ± 7.8 | 28.1 ± 2.0 |
| OP3 | **91.7 ± 1.7** | **93.6 ± 0.4** | **96.8 ± 0.5** | **96.3 ± 0.4** | **97.8 ± 0.1** | **2.0 ± 0.1** | 0.2 ± 0.0 | 6.1 ± 1.5 | **4.3 ± 0.2** |
| SCALOR | 74.1 ± 1.2 | 87.6 ± 0.4 | 67.9 ± 1.1 | 66.7 ± 1.1 | 78.4 ± 1.0 | 20.7 ± 1.0 | 0.8 ± 0.0 | **4.4 ± 0.4** | 14.0 ± 0.1 |

represents them independently by starting from an initial guess of the segmentation of the entire frame, and subsequently iteratively refines it using the refinement network $f$ (Marino et al., 2018). In each refinement step $m$, the image is represented by $K$ slots with latent representations $\mathbf{z}_{m,k}$. OP3 applies IODINE to each frame $x_t$ to extract latent representations $\mathbf{z}_{t,m,k}$, which are subsequently processed by a dynamics network $d$ (see Fig. 1e), which models both the individual dynamics of each slot $k$ as well as the pairwise interaction between all combinations of slots, aggregating them into a prediction of the posterior parameters for the next time step $t + 1$ for each slot $k$. For full details on IODINE and OP3, refer to Appendix C.4 and C.5, respectively.

**SCALable Object-oriented Representation (SCALOR)** (Jiang et al., 2020) is a spatial transformer-based model that factors scenes into background and multiple foreground objects, which are tracked throughout the sequence. Frames are encoded using a convolutional LSTM $f$. In the proposal-rejection phase, the current frame $t$ is divided into $H \times W$ grid cells. For each grid cell a object latent variable $\mathbf{z}_{t,h,w}$ is proposed, that is factored into existence, pose, depth and appearance parameters. Subsequently, proposed objects that significantly overlap with a propagated object are rejected. In the propagation phase, per object GRUs are updated for all objects present in the scene. Additionally, SCALOR has a background module to encode the background and its dynamics. Frame reconstructions are rendered using a background decoder and foreground STNs for object masks and appearance. For full details on SCALOR, refer to Appendix C.6.

## 4 RESULTS

We start with a summary of our overall results across the three datasets and four models (Table 2) before analyzing more specific challenging scenarios using variants of the VMDS dataset.

We first ask whether tracking could emerge automatically in an image-based model like MONET, which may produce consistent slot assignments through its learned object-slot assignment. This is not the case: MONET exhibits poor tracking performance (Table 2). While MONET correctly finds and segments objects, it does not assign them to consistent slots over time (Fig. E.2). In the following, we will thus focus on the video models: VIMON, TBA, OP3 and SCALOR.

**SpMOT**. All models perform tracking well on SpMOT with the exception of one training run of TBA with poor results leading to high standard deviation (cp. best TBA model: 89.8% MT; Table E.1). SCALOR outperforms the other models on the detection and tracking metrics MD and MT, while VIMON exhibits the highest MOTP, highlighting its better segmentation performance on SpMOT.

**VOR**. TBA is not applicable to VOR due to the dynamic background which cannot be resolved using background subtraction. VIMON and OP3 show similarly good performance on detection (MD) and

segmentation (MOTP), while VIMON outperforms OP3 on the tracking metrics MOTA and MT. OP3 accumulates a high number of false positives leading to a low MOTA due to the splitting of objects into multiple masks as well as randomly segmenting small parts of the background (Fig. E.4). In contrast, SCALOR has almost no false positives or ID switches, but accumulates a high number of misses leading to a poor MOTA. It often segments two objects as one that are occluding each other in the first frame, which is common in VOR due to the geometry of its scenes (Fig. F.11, last row).

**VMDS**. OP3 outperforms the other models on VMDS, on which TBA performs poorly, followed by SCALOR, which again accumulates a high number of misses. We will analyze the models on VMDS qualitatively and quantitatively in more detail in the following.

**Accumulation of evidence over time.** Recognition and tracking of objects should improve if models can exploit prior knowledge about the objects in the scene from previous video frames. To test whether the models exploit such knowledge, we evaluate their MOTA performance on VMDS after warm starting with up to 10 frames which are not included in evaluation (Fig. 2). Note that the models were trained on sequences of length 10, but are run for 20 frames in the case of a warm start of 10 frames. The performance of VIMON improves with longer warm starts, showing that the GRU accumulates evidence over time. TBA, in contrast, does not use temporal

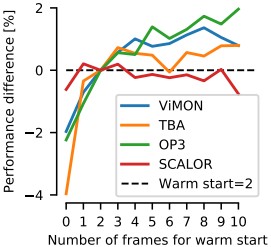
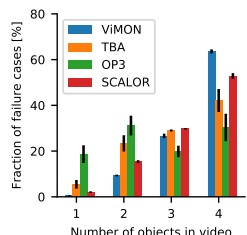

Figure 2: MOTA on frames 11–20 of the VMDS test set with warm starts of 1–10 frames (0 = no warm start). Difference to performance of warm start = 2 shown.

Figure 3: Distribution of failure cases dependent on number of objects in VMDS videos. Mean of three training runs. Error bars: standard deviation.

information beyond 2–3 frames, while SCALOR's performance slightly drops after 3 frames. OP3 appears to most strongly rely on past information and is able to integrate information over longer time scales: its performance does not even saturate with a warm start of 10 frames. However, the effect for all models is rather small.

**Challenging scenarios for different models.** *The number of objects in the sequence* matters for VIMON, TBA and SCALOR: more objects increase the number of failure cases (Fig. 3). In contrast, OP3 does not exhibit this pattern: it accumulates a higher number of false positives (FPs) in videos with fewer (only one or two) objects (Fig. E.1), as it tends to split objects into multiple slots if fewer objects than slots are present.

*Occlusion* leads to failure cases for all models (Fig. 4a–b). Partial occlusion can lead to splitting of objects into multiple slots (Fig. 4a). Objects that reappear after full occlusion are often missed when only a small part of them is visible (Fig. 4a). In particular, SCALOR tends to segment two objects as one when they overlap while entering the scene, leading to a high number of misses.

*Color* of the object is important. TBA often misses dark objects (Fig. 4b). In contrast, VIMON, OP3 and SCALOR struggle with scenes that feature objects of similar colors as well as objects that have similar colors to the background (Fig. 4c,e).

*False positives* are more prevalent for OP3 and TBA than for VIMON and SCALOR (Table 2). FPs of OP3 are due to objects split in multiple masks (Fig. 4a) and random small parts of the background being individually segmented (Fig. 4e), while TBA tends to compose larger objects using multiple smaller, simpler components (Fig. 4d).

**Challenge sets.** Based on the challenging scenarios identified above, we design multiple 'challenge sets': videos featuring *(1)* heavy occlusion, *(2)* objects with same colors, *(3)* only small objects and *(4)* only large objects (Fig. 5, top). For details, see Appendix (B.1.1).

Occlusion reduces performance of all models compared with the i.i.d. sampled VMDS test set, albeit to different degrees (Fig. 5; for absolute performance see Table E.2). OP3 is more robust to occlusion than the other models.

Tracking objects with the same color is challenging for all models (Fig. 5). In particular, OP3 appears to rely on object color as a way to separate objects.

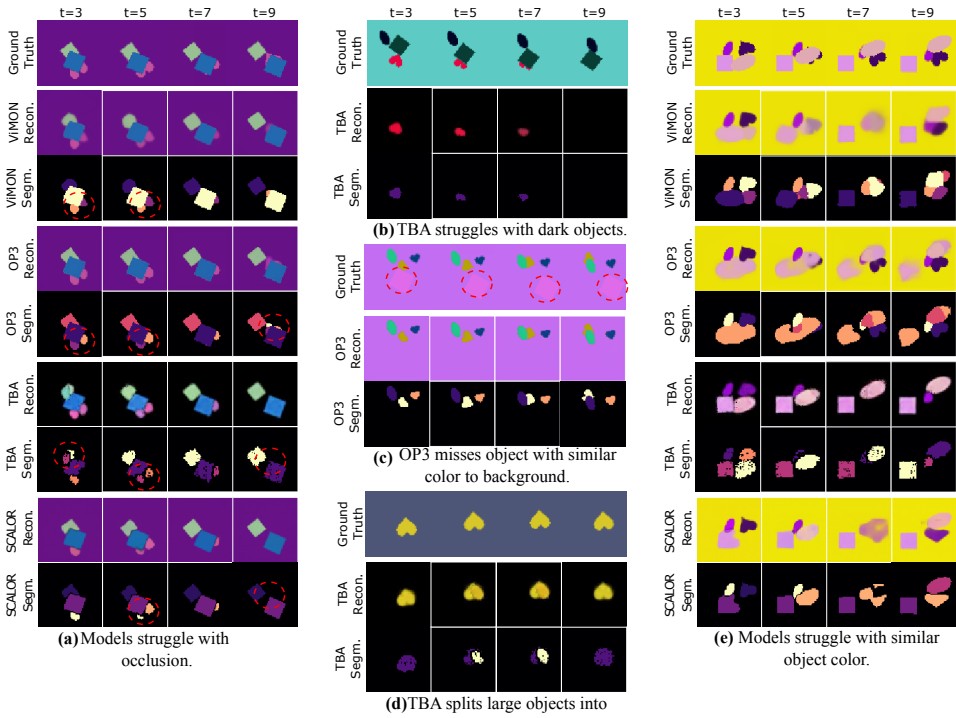

Figure 4: Example failure cases for all models on VMDS. Segmentation masks are binarized and color-coded to signify slot assignment.

OP3, VIMON and SCALOR are not sensitive to object size (Fig. 5). They exhibit only slightly decreased performance on the large objects test set, presumably because large objects cause more occlusion (Fig. 5). TBA shows increased performance on small objects but performs poorly on the large objects set.

**Out-of-distribution test sets.** Next, we assess generalization to out-of-distribution (o.o.d.) changes in object appearance that are not encountered during training. In the training set of VMDS, object color, size and orientation are constant throughout a video. To test o.o.d. generalization, we evaluate models trained on VMDS on three datasets that feature unseen object transformations (Fig. 6 and Table E.3): continuous changes in object **color** or **size** as well as continuous **rotation** around the object's centroid while moving. For details, see Appendix B.1.2.

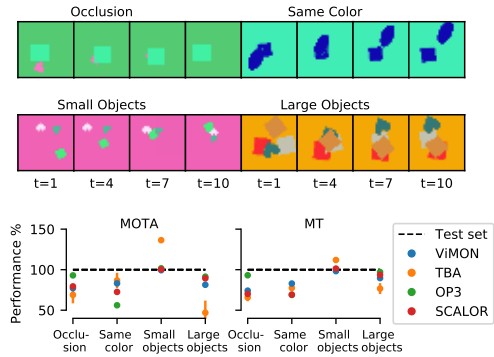

Figure 5: Performance on challenge sets relative to performance on VMDS test set (100%).

Continuous changes in object size do not pose a serious problem to TBA, OP3 and SCALOR, while VIMON's performance drops (Fig. 6). Surprisingly, continuous color changes of objects do not impact the performance of any model. Tracking performance of VIMON drops significantly for rotated objects, while OP3 and SCALOR are affected less. TBA's tracking performance is not as strongly influenced by object rotation (for absolute values, see Table E.3).

**Stability of training and runtime.** TBA and SCALOR train faster and require less memory than OP3 and VIMON (see Table E.4 for details). However, some training runs converge to suboptimal minima for TBA. Training OP3 is sensitive to the learning rate and unstable, eventually diverging in almost all experiments. Interestingly, it often reached its best performance prior to divergence. VIMON and TBA are less sensitive to hyper-parameter settings in our experiments. For a more detailed analysis of the runtime, see Appendix E.2

## 5 DISCUSSION

Our experimental results provide insights into the inductive biases and failure cases of object-centric models that were not apparent from their original publications. Despite the positive results shown in each of the papers for the evaluated methods, a controlled, systematic analysis demonstrates that they do not convincingly succeed at tracking, which is fundamentally what object-centric video methods should enable.

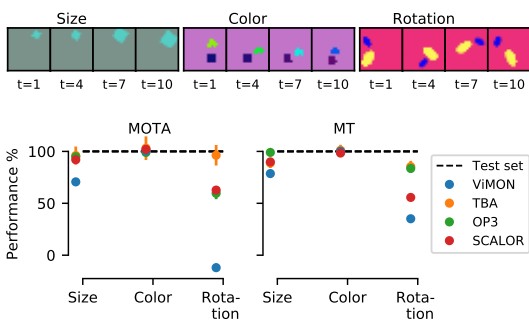

Figure 6: Performance on out-of-distribution sets relative to VMDS test set (100%).

TBA has a significantly lower MOTP than the other models on all datasets, suggesting that the simple rendering-based decoder using a fixed **template** might be less suitable to generate accurate segmentation masks (see also Fig. F.5 and Fig. F.4) compared to the VAE-based approaches of VIMON, OP3 and SCALOR.

Handling **occlusion** of objects during the video is a key property object-centric representations should be capable of. Qualitatively and quantitatively, OP3 is more robust to occlusion than the other models, suggesting that its dynamics network which models interactions between objects is currently most successful at modeling occluded objects. Surprisingly, TBA and SCALOR, which explicitly encode depth, do not handle occlusion more gracefully than VIMON, whose much simpler architecture has no explicit way of dealing with depth. Moving forward, occlusion handling is a key component that object-centric video models need to master, which can be addressed by either equipping the model with a potent interaction module, that takes pairwise interaction between objects (including occlusion) into account, similar to OP3's dynamics model, or ensuring that the depth reasoning of the models works as intended, which may be preferable, as explained below.

All models struggle with detecting objects that have similar **color** as the background (for TBA: dark objects, since background is removed and set to black in a pre-processing step). Color is a reliable cue to identify objects in these datasets. However, the auto-encoding objective incurs little extra loss for missing objects with similar color as the background and, thus, the models appear to not to learn to properly reconstruct them. In order to scale to data with more visual complexity, one might want to replace the pixel-wise reconstruction with for instance a loss based in feature space in order to focus more on reconstructing semantic content rather than high-frequency texture, as is done when using perceptual loss functions (Gatys et al., 2015; Hou et al., 2017) or by using contrastive learning (Kipf et al., 2020). Furthermore, the models – particularly so OP3 – struggle with separating objects of similar colors from each other. This result hints at a mismatch between the intuitions motivating these models and what the models actually learn: it should be more efficient in terms of the complexity of the latent representation to decompose two objects – even of similar colors – into two masks with simple shapes, rather than encoding the more complicated shape of two objects simultaneously in one slot. However, since none of the models handle occlusion with amodal segmentation masks (i.e. including the occluded portion of the object) successfully, they learn to encode overly complex (modal) mask shapes. As a consequence, they tend to merge similarly colored objects into one slot. This result suggests that resolving the issues surrounding the depth reasoning in combination with amodal segmentation masks would enable much more compact latents and could also resolve the merging of similarly colored objects.

A major difference between models is the **spatial transformer** based model formulation of TBA and SCALOR, compared to VIMON and OP3, which operate on image-sized masks. The parallel processing of objects and the processing of smaller bounding boxes renders training TBA and SCALOR to be significantly faster and more memory efficient, enabling them to scale to a larger number of objects. On the downside, the spatial transformer introduces its own complications. TBA depends strongly on its prior on object size and performs well only when this prior fits the data well as well as when the data contains little variation in object sizes, as in SpMOT (Table 2). However, it is not able to handle VMDS and its larger variation in object sizes and shapes. SCALOR performs tracking well in scenes where objects are clearly separated, but struggles to separate objects that partially occlude each other when entering the scene. This difficulty is caused by its discovery

mechanism which can propose at most one bounding box per grid cell, leading to a high number of misses on datasets which feature significant occlusion (VOR and VMDS). Unfortunately, simply increasing the number of proposals does not provide a simple solution, as SCALOR's performance is sensitive to properly tweaking the number of proposals.

Choosing a class of models is therefore dependent on the dataset one wants to apply it to as well as the computational resources at one's disposal. Datasets that feature a high number of objects (>10) that are well separated from each other make a method like SCALOR, which can process objects in parallel, advisable. On datasets with a lower number of objects per scene which feature heavy occlusion, methods like OP3 and ViMON will likely achieve better results, but require a high computational budget for training.

In conclusion, our analysis shows that none of the models solve the basic challenges of tracking even for relatively simple synthetic datasets. Future work should focus on developing robust mechanisms for reliably handling depth and occlusion, additionally combining the transformer-based efficiency of TBA and SCALOR with the stable training of VIMON and the interaction model of OP3. The key open challenges for scaling these models to natural videos include their computational inefficiency, complex training dynamics, as well as over-dependence on simple appearance cues like color.

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

# Supplementary Material for:
# Benchmarking Unsupervised Object Representations for Video Sequences

In this supplementary document, we first discuss the metrics used (Section A) and describe the data generation process (Section B). We then describe the methods MONet, ViMON, TBA, IODINE, OP3 and SCALOR (Section C). Section D contains information regarding the implementation details and training protocols. Finally, we provide additional qualitative and quantitative experimental results in Section E.

## A    Evaluation Protocol Details

We quantitatively evaluate all models on three datasets using the standard CLEAR MOT metrics (Bernardin & Stiefelhagen, 2008). Our evaluation protocol is adapted from the multi-object tracking (MOT) challenge (Milan et al., 2016), a standard computer vision benchmark for supervised object tracking. In particular, we focus on the metrics provided by the py-motmetrics package[2].

### A.1    Mapping

In each frame, object predictions of each model in the form of binary segmentation masks are mapped to the ground truth object segmentation masks. We require that each pixel is uniquely assigned to at most one object in the ground truth and the predictions, respectively. Matching is based on the intersection over union (IoU) between the predictions and the ground truth masks (Voigtlaender et al., 2019). A valid correspondence between prediction and object has to exceed a threshold in IoU of 0.5. Predictions that are not mapped to any ground truth mask are classified as false positives (FPs). Ground truth objects that are not matched to any prediction are classified as misses. Following (Bernardin & Stiefelhagen, 2008), ground truth objects that are mapped to two different hypothesis IDs in subsequent frames are classified as ID switches for that frame.

### A.2    MOT Metrics

**MOT Accuracy (MOTA)** measures the fraction of all failure cases, i.e. false positives (FPs), misses and ID switches compared to total number of objects present in all frames. **MOT Precision (MOTP)** measures the total accuracy in position for matched object hypothesis pairs, relative to total number of matches made. We use percentage Intersection over Union (IoU) of segmentation masks as the accuracy in position for each match. **Mostly Tracked (MT)** is the ratio of ground truth objects that have been tracked for at least 80% of their lifespan.(i.e. 80% of the frames in which they are visible). MT as implemented by py-motmetrics counts trajectories of objects as correctly tracked even if ID switches occur. We use a strictly more difficult definition of MT that counts trajectories with ID switches as correctly detected but not correctly tracked. Consequently, we add the **Mostly Detected (MD)** measure which does not penalize ID switches. **Match**, **Miss**, **ID Switches (ID S.)** and **FPs** are reported as the fraction of the number of occurrences divided by the total number of object occurrences.

$$\text{MOTA} = 1 - \frac{\sum_{t=1}^{T} \text{M}_t + \text{FP}_t + \text{IDS}_t}{\sum_{t=1}^{T} \text{O}_t} \tag{1}$$

where $\text{M}_t$, $\text{FP}_t$, and $\text{IDS}_t$ are the number of misses, false positives and ID switches, respectively, for time $t$, and $\text{O}_t$ is the number of objects present in frame $t$. Note that MOTA can become negative, since the number of FPs is unbounded.

$$\text{MOTP} = \frac{\sum_{t=1}^{T} \sum_{i=1}^{I} d_t^i}{\sum_{t=1}^{T} c_t} \tag{2}$$

where $d_t^i$ is the total accuracy in position for the $i^{\text{th}}$ matched object-hypothesis pair measured in IoU between the respective segmentation masks and $c_t$ is the number of matches made in frame $t$.

---

[2]https://pypi.org/project/motmetrics/

Note that we exclude the background masks for VIMON and OP3 before evaluating tracking based on IoU. The Video Object Room (VOR) dataset can contain up to three background segments, namely the floor and up to two wall segments. In order to exclude all background slots regardless of whether the model segments the background as one or as multiple masks, we remove all masks before the tracking evaluation that have an IoU of more than 0.2 with one of the ground truth background masks; we empirically tested that this heuristic is successful in removing background masks regardless of whether the models segments it as one or as three separate ones.

# B    DATASET GENERATION DETAILS

## B.1    VIDEO MULTI-DSPRITES (VMDS)

The Multi-DSprites Video dataset consists of 10-frame video sequences of 64×64 RGB images with multiple moving sprites per video. In order to test temporal aggregation properties of the models, the test set contains 20 frame-long sequences. Each video contains one to four sprites following the dataset proposed in (Burgess et al., 2019) that move independently of each other and might partially or fully occlude one another. The sprites are sampled uniformly from the dSprites dataset (Matthey et al., 2017) and colored with a random RGB color. The background is uniformly colored with a random RGB color. Random trajectories are sampled per object by drawing $x$ and $y$ coordinates from a Gaussian process with squared exponential covariance kernel $\text{cov}[x_s, x_t] = \exp[-(x_s - x_t)^2/(2\tau^2)]$ and time constant $\tau = 10$ frames, and then shifted by an initial $(x, y)$-position of the sprite centroid, which is uniformly sampled from $[10, 54]$ to ensure that the object is within the image boundaries. Trajectories that leave these boundaries are rejected. In occlusion scenarios, larger objects are always in front of smaller objects to disambiguate prediction of occlusion. The training set consists of 10,000 examples whereas the validation set as well as the test set contain 1,000 examples each. Additionally, we generated four challenge sets and three out-of-distribution test sets for VMDS that contain specifically challenging scenarios. Each test set consists of 1,000 videos of length 10 frames, which we describe in the following.

### B.1.1    VMDS CHALLENGE SETS

**Occlusion test set.** In each video, one or more objects are heavily occluded and thus often are not visible at all for a few frames. This is ensured by sampling object trajectories that cross path, i.e. at least in one video frame, two objects are centered on the same pixel. The time step and spatial position of occlusion is sampled randomly. Object trajectories are sampled independently as described above and then shifted such that they are at the sampled position of occlusion at time $t$. Videos contain two to four sprites (Fig. 5), since at least two objects are necessary for occlusion.

**Small Objects.** Videos contain one to four sprites with all sprites being of the smallest size present in the original dSprites (Matthey et al., 2017) dataset (Fig. 5). Other than that, it follows the generation process of the regular training and test set.

**Large Objects.** Videos contain one to four sprites with all sprites being of the largest size present in the original dSprites (Matthey et al., 2017) dataset (Fig. 5). Other than that, it follows the generation process of the regular training and test set.

**Same Color.** Videos contain two to four sprites which are identically colored with a randomly chosen color. Other than that, it follows the generation process of the regular training and test set (Fig. 5).

### B.1.2    VMDS OUT-OF-DISTRIBUTION TEST SETS

**Rotation test set**. Sprites rotate around their centroid while moving. The amount of rotation between two video frames is uniformly sampled between 5 and 40 degrees, and is constant for each object over the course of the video. Direction of rotation is chosen randomly. Rotation is not included as a transformation in the training set (Fig. 6).

**Color change test set.** Sprites change their color gradually during the course of the video. The initial hue of the color is chosen randomly as well as the direction and amount of change between two frames, which stays the same for each object over the course of the video. Saturation and value of the color are kept constant. Color changes are not part of the training set (Fig. 6).

**Size change test set.** Sprites change their size gradually during the course of the video. The original dSprites dataset (Matthey et al., 2017) contains six different sizes per object. For each object, its size is sampled as either the smallest or largest in the first frame as well as a random point in time, at which it starts changing its size. At this point in time, it will either become larger or smaller, respectively, increasing or decreasing each frame to the next larger or smaller size present in the original dSprites dataset, until the largest or smallest size is reached. Size changes are not part of the training set (Fig. 6).

### B.2 SPRITES-MOT (SPMOT)

Sprites-MOT, originally introduced by (He et al., 2019), consists of video sequences of length 20 frames. Each frame is a 128×128 RGB image. It features multiple sprites moving linearly on a black background. The sprite can have one of four shapes and one of six colors. For more information, refer to the original paper (He et al., 2019). We generate a training set consisting of 9600 examples, validation set of 384 samples and test set of 1,000 examples using the author-provided public codebase[3]. However, instead of using the default setting of 20 frames per sequence, we instead generate sequences of length 10, in order to facilitate comparison to the other datasets in our study which have only 10 frames per sequence.

Frames are downsampled to a resolution of 64×64 for training VIMON, OP3 and SCALOR.

### B.3 VIDEO OBJECTS ROOM (VOR)

We generate a video dataset based on the static Objects Room dataset (Greff et al., 2019), with sequences of length 10 frames each at a resolution of 128×128. This dataset is rendered with OpenGL using the gym-miniworld[4] reinforcement learning environment. It features a 3D room with up to four static objects placed in one quadrant of the room, and a camera initialized at the diagonally opposite quadrant. The objects are either static cubes or spheres, assigned one of 6 colors and a random orientation on the ground plane of the room. The camera then follows one of five trajectories moving towards the objects, consisting of a small fixed distance translation and optional small fixed angle of rotation each time step. The wall colors and room lighting are randomized, but held constant throughout a sequence. The training set consists of 10,000 sequences whereas the validation set and the test set contain 1,000 sequences each.

Frames are downsampled to a resolution of 64×64 for training VIMON, OP3 and SCALOR.

## C METHODS

In this section we describe the various methods in a common mathematical framework. For details about implementation and training, please refer to Section D.

### C.1 MONET

Multi-Object-Network (MONET) (Burgess et al., 2019) is an object-centric representation model designed for static images. It consists of a recurrent attention network that sequentially extracts attention masks of individual objects and a variational autoencoder (VAE) (Kingma & Welling, 2014) that reconstructs the image region given by the attention mask in each processing step.

**Attention Network:** The attention network is a U-Net (Ronneberger et al., 2015) parameterized by $\psi$. At each processing step $k$, the attention network receives the full image $\mathbf{x} \in [0, 1]^{H \times W \times 3}$ as input together with the scope variable $\mathbf{s}_k \in [0, 1]^{H \times W}$. The scope $\mathbf{s}_k$ keeps track of the regions of the image that haven't been attended to in the previous processing steps and thus remain to be explained. The attention network outputs a soft attention mask $\mathbf{m}_k \in [0, 1]^{H \times W}$ and the updated scope with the current mask subtracted:

---

[3] https://github.com/zhen-he/tracking-by-animation
[4] https://github.com/maximecb/gym-miniworld

$$\mathbf{m}_k = \mathbf{s}_{k-1}\alpha_\psi(\mathbf{x}, s_{k-1}) \tag{3}$$
$$\mathbf{s}_{k+1} = \mathbf{s}_k(1 - \alpha_\psi(\mathbf{x}, \mathbf{s}_k)) \tag{4}$$

where $\alpha_\psi(\mathbf{x}, \mathbf{s}_k) \in [0,1]^{H \times W}$ is the output of the U-net and $s_0 = \mathbf{1}$. The attention mask for the last slot is given by $\mathbf{m}_K = \mathbf{s}_{K-1}$ to ensure that the image is fully explained, i.e. $\sum_{k=1}^K \mathbf{m}_k = \mathbf{1}$.

**VAE:** The VAE consists of an encoder $g : [0,1]^{H \times W \times 3} \times [0,1]^{H \times W} \rightarrow \mathbb{R}^{L \times 2}$ and a decoder $h : \mathbb{R}^L \rightarrow [0,1]^{H \times W \times 3} \times [0,1]^{H \times W}$ which are two neural networks parameterized by $\phi$ and $\theta$, respectively. The VAE encoder receives as input the full image $\mathbf{x}$ and the attention mask $\mathbf{m}_k$ and computes $(\boldsymbol{\mu}_k, \log \boldsymbol{\sigma}_k)$, which parameterize the Gaussian latent posterior distribution $q_\phi(\mathbf{z}_k|\mathbf{x}, \mathbf{m}_k) = \mathcal{N}(\boldsymbol{\mu}_k, \boldsymbol{\sigma}_k I)$. Using the reparametrization trick (Kingma & Welling, 2014), $\mathbf{z}_k \in \mathbb{R}^L$ is sampled from the latent posterior distribution. $\mathbf{z}_k$ is decoded by the VAE decoder into a reconstruction of the image component $\widehat{\mathbf{x}}_k \in [0,1]^{H \times W \times 3}$ and mask logits, which are used to compute the reconstruction of the mask $\widehat{\mathbf{m}}_k \in [0,1]^{H \times W}$ via a pixelwise softmax across slots. The reconstruction of the whole image is composed by summing over the K masked reconstructions of the VAE: $\widehat{\mathbf{x}} = \sum_{k=1}^K \widehat{\mathbf{m}}_k \odot \widehat{\mathbf{x}}_k$.

**Loss:** MONET is trained end-to-end with the following loss function:

$$L(\phi; \theta; \psi; \mathbf{x}) = -\log \sum_{k=1}^K \mathbf{m}_k p_\theta(\mathbf{x}|\mathbf{z}_k) + \beta D_{\mathrm{KL}}(\prod_{k=1}^K q_\phi(\mathbf{z}_k|\mathbf{x}, \mathbf{m}_k) \| p(\mathbf{z}))$$
$$+ \gamma \sum_{k=1}^K D_{\mathrm{KL}}(q_\psi(\mathbf{m}_k|\mathbf{x}) \| p_\theta(\mathbf{m}_k|\mathbf{z}_k)) \tag{5}$$

where $p_\theta(\mathbf{x}|\mathbf{z}_k)$ is the Gaussian likelihood of the VAE decoder and $\mathbf{z}_k \in \mathbb{R}^L$ is the latent representation of slot $k$.

The first two loss terms are derived from the standard VAE objective, the Evidence Lower BOund (ELBO) (Kingma & Welling, 2014), i.e. the negative log-likelihood of the decoder and the Kullback–Leibler divergence between the unit Gaussian prior $p(\mathbf{z}) = \mathcal{N}(\mathbf{0}, I)$ and the latent posterior distribution $q_\phi(\mathbf{z}_k|\mathbf{x}, \mathbf{m}_k)$ factorized across slots. Notably, the decoder log-likelihood term $p_\theta(\mathbf{x}|\mathbf{z}_k)$ constrains only the reconstruction within the mask, since it is weighted by the mask $\mathbf{m}_k$. Additionally, as a third term, the Kullback–Leibler divergence of the attention mask distribution $q_\psi(\mathbf{m}_k|\mathbf{x})$ with the VAE mask distribution $p_\theta(\widehat{\mathbf{m}}_k|\mathbf{z}_k)$ is minimized, to encourage the VAE to learn a good reconstruction of the masks.

## C.2    VIDEO MONET

We propose an extension of MONET (Burgess et al., 2019), called Video MONet (VIMON), which accumulates evidence over time about the objects in the scene (Fig. C.1).

VIMON processes a video recurrently by reconstructing one frame at a time and predicting the next frame of the video. The processing of each frame follows a logic similar to MONET with some notable differences. In the following, we use $t$ to indicate the time step in the video and $k$ to indicate the processing step within one video frame.

**Attention Network:** The attention network of VIMON outputs an attention mask $\mathbf{m}_{t,k} \in [0,1]^{H \times W}$ in each step $k$ conditioned on the full frame $\mathbf{x}_t \in [0,1]^{H \times W \times 3}$, the scope $\mathbf{s}_{t,k} \in [0,1]^{H \times W}$ and additionally the mask $\widehat{\mathbf{m}}_{t,k} \in [0,1]^{H \times W}$ that was predicted by the

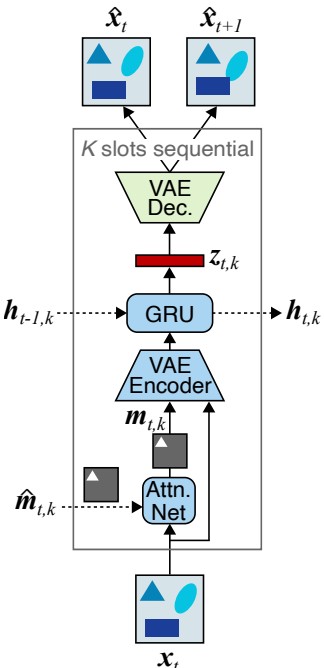

Figure C.1: **VIMON.** Attention network followed by VAE encoder and GRU computes latent $\mathbf{z}_{t,k}$.

VAE in the previous time step, in order to provide it with information about which object it should attend to in this specific slot $k$.

$$\mathbf{m}_{t,k} = \mathbf{s}_{t,k-1}\alpha_\psi(\mathbf{x}_t, \mathbf{s}_{t,k-1}, \widehat{\mathbf{m}}_{t,k}) \tag{6}$$

**VAE:** The VAE of VIMON consists of an encoder $g(\mathbf{x}_t, \mathbf{m}_{t,k}; \phi)$ and a decoder $h(\mathbf{z}_{t,k}; \theta)$. In contrast to MONET, the encoder in VIMON is followed by a gated recurrent unit (GRU) (Cho et al., 2014) with a separate hidden state $h_{t,k}$ per slot $k$. Thus, the GRU aggregates information over time for each object separately. The GRU outputs $(\boldsymbol{\mu}_{t,k}, \log \boldsymbol{\sigma}_{t,k})$ which parameterize the Gaussian latent posterior distribution $q_\phi(\mathbf{z}_{t,k}|\mathbf{x}_t, \mathbf{m}_{t,k})$ where $\mathbf{z}_{t,k} \in \mathbb{R}^L$ is the latent representation for slot $k$ at time $t$:

$$\mathbf{z}'_{t,k} = g(\mathbf{x}_t, \mathbf{m}_{t,k}; \phi) \tag{7}$$

$$(\boldsymbol{\mu}_{t,k}, \log \boldsymbol{\sigma}_{t,k}), \mathbf{h}_{t,k} = f(GRU(\mathbf{z}'_{t,k}, \mathbf{h}_{t-1,k}))) \tag{8}$$

$$q_\phi(\mathbf{z}_{t,k}|\mathbf{x}_t, \mathbf{m}_{t,k}) = \mathcal{N}(\boldsymbol{\mu}_{t,k}, \boldsymbol{\sigma}_{t,k}I) \qquad \forall t, k \tag{9}$$

where $g$ is the VAE encoder and $f$ is a linear layer. The latent representation $\mathbf{z}_{t,k}$ is sampled from the latent posterior distribution using the reparametrization trick (Kingma & Welling, 2014). Subsequently, $\mathbf{z}_{t,k}$ is linearly transformed into $\widehat{\mathbf{z}}_{t+1,k}$ via a learned transformation $\mathbf{A} \in \mathbb{R}^{L \times L}$: $\widehat{\mathbf{z}}_{t+1,k} = \mathbf{A}\mathbf{z}_{t,k}$ with $\widehat{\mathbf{z}}_{t+1,k}$ being the predicted latent code for the next time step $t+1$. Both $\mathbf{z}_{t,k}$ and $\widehat{\mathbf{z}}_{t+1,k}$ are decoded by the shared VAE decoder $h_\theta$ into a reconstruction of the image $\widehat{\mathbf{x}}_{t,k} \in [0,1]^{H \times W \times 3}$ and a reconstruction of the mask $\widehat{\mathbf{m}}_{t,k} \in [0,1]^{H \times W}$ as well as $\widehat{\mathbf{x}}_{t+1,k}$ and $\widehat{\mathbf{m}}_{t+1,k}$, respectively.

**Loss:** VIMON is trained in an unsupervised fashion with the following objective adapted from the MONET loss (Eq. (5)) for videos. To encourage the model to learn about object motion, we include a prediction objective in the form of a second decoder likelihood on the next-step prediction $p_\theta(\mathbf{x}_{t+1}|\widehat{\mathbf{z}}_{t+1,k})$ and an additional mask loss term, which encourages the predicted VAE mask distribution $p_\theta(\widehat{\mathbf{m}}_{t+1,k}|\widehat{\mathbf{z}}_{t+1,k})$ to be close to the attention mask distribution $q_\psi(\mathbf{m}_{t+1,k}|\mathbf{x}_{t+1})$ of the next time step for each slot $k$:

$$L(\phi; \theta; \psi; \mathbf{x}) = \sum_{t=1}^{T} L_{\text{negLL}} + \beta L_{\text{prior}} + \gamma L_{\text{mask}}$$

$$L_{\text{negLL}} = -(\log \sum_{k=1}^{K} \mathbf{m}_{t,k} p_\theta(\mathbf{x}_t|\mathbf{z}_{t,k}) + \log \sum_{k=1}^{K} \mathbf{m}_{t+1,k} p_\theta(\mathbf{x}_{t+1}|\widehat{\mathbf{z}}_{t+1,k}))$$

$$L_{\text{prior}} = D_{\text{KL}}(\prod_{k=1}^{K} q_\phi(\mathbf{z}_{t,k}|\mathbf{x}_t, \mathbf{m}_{t,k})\|p(\mathbf{z}))$$

$$L_{\text{mask}} = \sum_{k=1}^{K} D_{\text{KL}}(q_\psi(\mathbf{m}_{t,k}|\mathbf{x}_t)\|p_\theta(\mathbf{m}_{t,k}|\mathbf{z}_{t,k})) + D_{\text{KL}}(q_\psi(\mathbf{m}_{t+1,k}|\mathbf{x}_{t+1})\|p_\theta(\mathbf{m}_{t+1,k}|\widehat{\mathbf{z}}_{t+1,k}))$$

### C.3 TRACKING BY ANIMATION

Tracking by Animation (TBA) (He et al., 2019) is a spatial transformer-based attention model which uses a simple 2D rendering pipeline as the decoder. Objects are assigned tracking templates and pose parameters by a tracker array, such that they can be reconstructed in parallel using a renderer based on affine spatial transformation (Fig. C.2). In contrast to VIMON, TBA uses explicit parameters to encode the position, size, aspect ratio and occlusion properties for each slot. Importantly, TBA is designed for scenes with static backgrounds, and preprocesses sequences using background subtraction (Bloisi & Iocchi, 2012) before they are input to the tracker array.

**Tracker Array:** TBA uses a tracker array to output a latent representation $\mathbf{z}_t \in \mathbb{R}^{L \times K}$ at time $t$ using a feature extractor $f(\mathbf{x}_t; \psi)$ and a recurrent 'state update', where $\mathbf{c}_t \in \mathbb{R}^{M \times N \times C}$ is a convolutional feature representation. The convolutional feature and latent representation have far fewer elements than $\mathbf{x}_t$, acting as a bottleneck:

$$\mathbf{c}_t = f(\mathbf{x}_t; \psi), \tag{10}$$
$$\mathbf{h}_{t,k} = RAT(\mathbf{h}_{t-1,k}, \mathbf{c}_t; \pi), \tag{11}$$
$$\mathbf{z}_t = g(\mathbf{h}_t; \phi). \tag{12}$$

Though the state update could be implemented as any generic recurrent neural network block, such as an LSTM (Hochreiter & Schmidhuber, 1997) or GRU (Cho et al., 2014), TBA introduces a Reprioritized Attentive Tracking (RAT) block that uses attention to achieve explicit association of slots with similar features over time. Firstly, the previous tracker state $\mathbf{h}_{t-1,k}$ is used to generate key variables $\mathbf{k}_{t,k}$ and $\beta_{t,k}$:

$$\{\mathbf{k}_{t,k}, \widehat{\beta}_{t,k}\} = \mathbf{T}\mathbf{h}_{t-1,k}, \tag{13}$$
$$\beta_{t,k} = 1 + \ln(1 + \exp(\widehat{\beta}_{t,k})), \tag{14}$$

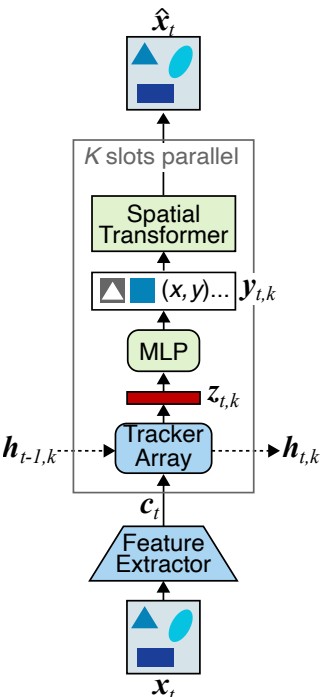

Figure C.2: **TBA**. Feature extractor CNN $f$ and tracker array $g$ to get latent $\mathbf{z}_{t,k}$. MLP $h$ outputs mid-level representation $\mathbf{y}_{t,k}$, and Spatial Transformer renders reconstruction.

where $\mathbf{T}$ is a learned linear transformation, $\mathbf{k}_{t,k} \in \mathbb{R}^S$ is the addressing key, and $\widehat{\beta}_{t,k} \in \mathbb{R}$ is an un-normalized version of a key strength variable $\beta_{t,k} \in (1, +\infty)$. This key strength acts like a temperature parameter to modulate the feature re-weighting, which is described in the following. Each feature vector in $\mathbf{c}_t$, denoted by $\mathbf{c}_{t,m,n} \in \mathbb{R}^S$, where $m \in \{1, 2, \dots, M\}$ and $n \in \{1, 2, \dots, N\}$ are the convolutional feature dimensions, is first used to get attention weights:

$$W_{t,k,m,n} = \frac{\exp(\beta_{t,k} Sim(\mathbf{k}_{t,k}, \mathbf{c}_{t,m,n}))}{\sum_{m',n'} \exp(\beta_{t,k} Sim(\mathbf{k}_{t,k}, \mathbf{c}_{t,m',n'}))}. \tag{15}$$

Here, $Sim$ is the cosine similarity defined as $Sim(\mathbf{p}, \mathbf{q}) = \mathbf{p}\mathbf{q}^\mathsf{T}/(\|\mathbf{p}\|\|\mathbf{q}\|)$, and $W_{t,k,m,n}$ is an element of the attention weight $\mathbf{W}_{t,k} \in [0,1]^{M \times N}$, satisfying $\sum_{m,n} W_{t,k,m,n} = 1$. Next, a read operation is defined as a weighted combination of all feature vectors of $\mathbf{c}_t$:

$$\mathbf{r}_{t,k} = \sum_{m,n} W_{t,k,m,n} \, \mathbf{c}_{t,m,n} \tag{16}$$

where $\mathbf{r}_{t,k} \in \mathbb{R}^S$ is the read vector, representing the associated input feature for slot $k$. Intuitively, for slots in which objects are present in the previous frame, the model can suppress the features in $\mathbf{r}_{t,k}$ that are not similar to the features of that object, helping achieve better object-slot consistency. On the other hand, if there are slots which so far do not contain any object, the key strength parameter allows $\mathbf{r}_{t,k}$ to remain similar to $\mathbf{c}_t$ facilitating the discovery of new objects.

The tracker state $\mathbf{h}_{t,k}$ of the RAT block is updated with an RNN parameterized by $\pi$, taking $\mathbf{r}_{t,k}$ instead of $\mathbf{c}_t$ as its input feature:

$$\mathbf{h}_{t,k} = RNN(\mathbf{h}_{t-1,k}, \mathbf{r}_{t,k}; \pi) \tag{17}$$

The RAT block additionally allows for sequential prioritization of trackers, which in turn allows only a subset of trackers to update their state at a given time step, improving efficiency. For full details on the reprioritization and adaptive computation time elements of the RAT block, please refer to the original paper (He et al., 2019).

**Mid-Level Representation:** The key feature of TBA is that each latent vector $\mathbf{z}_{t,k}$ is further decoded into a mid-level representation $\mathbf{y}_{t,k} = \{y_{t,k}^c, \mathbf{y}_{t,k}^l, \mathbf{y}_{t,k}^p, \mathbf{Y}_{t,k}^s, \mathbf{Y}_{t,k}^a\}$ corresponding to interpretable, explicit object properties, via a fully-connected neural network $h(\mathbf{z}_{t,k}; \theta)$ as follows:

$$\mathbf{y}_{t,k} = h(\mathbf{z}_{t,k}; \theta). \tag{18}$$

$h_\theta$ is shared by all slots, improving parameter efficiency. The different components of the mid-level representation are:

- Confidence $y_{t,k}^c \in [0, 1]$: Probability of existence of an object in that slot.

- Layer $\mathbf{y}_{t,k}^l \in \{0, 1\}^O$: One-hot encoding of the discretized pseudo-depth of the object relative to other objects in the frame. Each image is considered to be composed of $O$ object layers, where higher layer objects occlude lower layer objects and the background is the zeroth (lowest) layer. E.g., when $O = 4$, $\mathbf{y}_{t,k}^l = [0, 0, 1, 0]$ denotes the third layer. For simplicity and without loss of generality, we can also denote the same layer with its integer representation $y_{t,k}^l = 3$.

- Pose $\mathbf{y}_{t,k}^p = [\widehat{s}_{t,k}^x, \widehat{s}_{t,k}^y, \widehat{t}_{t,k}^x, \widehat{t}_{t,k}^y] \in [-1, 1]^4$: Normalized object pose for calculating the scale $[s_{t,k}^x, s_{t,k}^y] = [1 + \eta^x \widehat{s}_{t,k}^x, 1 + \eta^y \widehat{s}_{t,k}^y]$ and the translation $[t_{t,k}^x, t_{t,k}^y] = [\frac{W}{2}\widehat{t}_{t,k}^x, \frac{H}{2}\widehat{t}_{t,k}^y]$, where $\eta^x, \eta^y > 0$ are constants.

- Shape $\mathbf{Y}_{t,k}^s \in \{0, 1\}^{U \times V}$ and Appearance $\mathbf{Y}_{t,k}^a \in [0, 1]^{U \times V \times 3}$: Object template, with hyperparameters $U$ and $V$ typically set much smaller than the image dimensions $H$ and $W$. Note that the shape is discrete (for details, see below) whereas the appearance is continuous.

In the output layer of $h_\theta$, $y_{t,k}^c$ and $\mathbf{Y}_{t,k}^a$ are generated by the sigmoid function, $\mathbf{y}_{t,k}^p$ is generated by the tanh function, and $\mathbf{y}_{t,k}^l$ as well as $\mathbf{Y}_{t,k}^s$ are sampled from the Categorical and Bernoulli distributions, respectively. As sampling is non-differentiable, the Straight-Through Gumbel-Softmax estimator (Jang et al., 2017) is used to reparameterize both distributions so that backpropagation can still be applied.

**Renderer:** To obtain a frame reconstruction, the renderer scales and shifts $\mathbf{Y}_{t,k}^s$ and $\mathbf{Y}_{t,k}^a$ according to $\mathbf{y}_{t,k}^p$ via a Spatial Transformer Network (STN) (Jaderberg et al., 2015):

$$\mathbf{m}_{t,k} = STN(\mathbf{Y}_{t,k}^s, \mathbf{y}_{t,k}^p), \tag{19}$$

$$\widehat{\mathbf{x}}_{t,k} = STN(\mathbf{Y}_{t,k}^a, \mathbf{y}_{t,k}^p). \tag{20}$$

where $\mathbf{m}_{t,k} \in \{0, 1\}^D$ and $\widehat{\mathbf{x}}_{t,k} \in [0, 1]^{D \times 3}$ are the spatially transformed shape and appearance respectively. To obtain the final object masks $\widehat{\mathbf{m}}_{t,k}$, an occlusion check is performed by initializing $\widehat{\mathbf{m}}_{t,k} = y_{t,k}^c \mathbf{m}_{t,k}$, then removing the elements of $\widehat{\mathbf{m}}_{t,k}$ for which there exists an object in a higher layer. That is, for $k = 1, 2, \ldots, K$ and $\forall j \neq k$ where $y_{t,j}^l > y_{t,k}^l$:

$$\widehat{\mathbf{m}}_{t,k} = (\mathbf{1} - \mathbf{m}_{t,j}) \odot \widehat{\mathbf{m}}_{t,k}. \tag{21}$$

In practice, the occlusion check is sped up by creating intermediate 'layer masks', partially parallelizing the operation. Please see the original paper for more details (He et al., 2019). The final reconstruction is obtained by summing over the $K$ slots, $\widehat{\mathbf{x}}_t = \sum_{k=1}^{K} \widehat{\mathbf{m}}_{t,k} \odot \widehat{\mathbf{x}}_{t,k}$.

**Loss:** Learning is driven by a pixel-level reconstruction objective, defined as:

$$L(\phi; \psi; \pi; \theta; \mathbf{x}) = \sum_{t=1}^{T} \left( MSE(\widehat{\mathbf{x}}_t, \mathbf{x}_t) + \lambda \cdot \frac{1}{K} \sum_{k=1}^{K} s_{t,k}^x \, s_{t,k}^y \right), \tag{22}$$

where $MSE$ refers to the mean squared error and the second term penalizes large scales $[s_{t,k}^x, s_{t,k}^y]$ in order to make object bounding boxes more compact.

## C.4 IODINE

The Iterative Object Decomposition Inference NEtwork (IODINE) (Greff et al., 2019), similar to MONET (Burgess et al., 2019), learns to decompose a static scene into a multi-slot representation, in which each slot represents an object in the scene and the slots share the underlying format of the independent representations. In contrast to MONET, it does not recurrently segment the image using spatial attention, rather it starts from an initial guess of the segmentation of the whole image and iteratively refines it. Thus, the inference component of both models differ, while the generative component is the same.

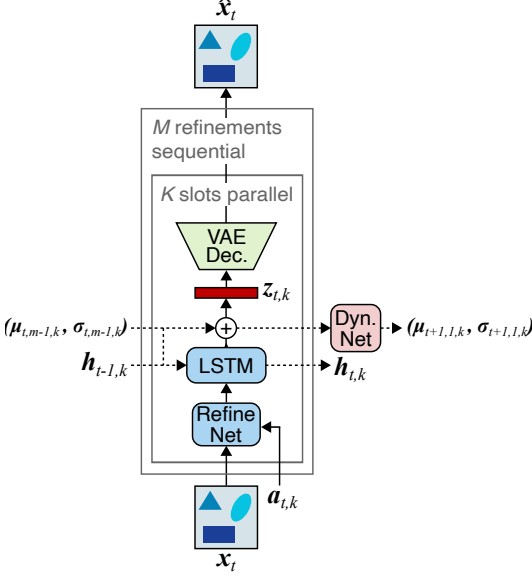

Figure C.3: **OP3**. Refinement network $f$ followed by LSTM and dynamics network $d$ compute latent $\mathbf{z}_{t,k}$.

**Iterative Inference.** As with MONET, IODINE models the latent posterior $q(\mathbf{z}_k|\mathbf{x})$ per slot $k$ as a Gaussian parameterized by $(\boldsymbol{\mu}_{m,k}, \boldsymbol{\sigma}_{m,k}) \in \mathbb{R}^{L \times 2}$. To obtain latent representations for independent regions of the input image, IODINE starts from initial learned posterior parameters $(\boldsymbol{\mu}_{1,k}, \boldsymbol{\sigma}_{1,k})$ and iteratively refines them using the refinement network $f_\phi$ for a fixed number of refinement steps $M$. $f_\phi$ consists of a convolutional neural network (CNN) in combination with an LSTM cell (Hochreiter & Schmidhuber, 1997) parameterized by $\phi$. In each processing step, $f_\phi$ receives as input the image $\mathbf{x} \in [0,1]^{H \times W \times 3}$, a sample from the current posterior estimate $\mathbf{z}_{m,k} \in \mathbb{R}^L$ and various auxiliary inputs $\mathbf{a}_k$, which are listed in the original paper (Greff et al., 2019). The posterior parameters are concatenated with the output of the convolutional part of the refinement network and together form the input to the refinement LSTM. The posterior parameters are additively updated in each step $m$ in parallel for all $K$ slots:

$$(\boldsymbol{\mu}_{m+1,k}, \boldsymbol{\sigma}_{m+1,k}) = (\boldsymbol{\mu}_{m,k}, \boldsymbol{\sigma}_{m,k}) + f_\phi(\mathbf{z}_{m,k}, \mathbf{x}, \mathbf{a}_k) \tag{23}$$

**Decoder**. In each refinement step $m$, the image is represented by $K$ latent representations $\mathbf{z}_{m,k}$. Similar to MONET, each $\mathbf{z}_{m,k}$ is independently decoded into a reconstruction of the image $\widehat{\mathbf{x}}_{m,k} \in [0,1]^{H \times W \times 3}$ and mask logits $\widetilde{\mathbf{m}}_{m,k}$, which are subsequently normalized by applying the softmax across slots to obtain the masks $\mathbf{m}_{m,k} \in [0,1]^{H \times W}$. The reconstruction of the whole image at each refinement step $m$ is composed by summing over the K masked reconstructions of the decoder: $\widehat{\mathbf{x}} = \sum_{k=1}^K \mathbf{m}_{m,k} \odot \widehat{\mathbf{x}}_{m,k}$.

**Training.** IODINE is trained by minimizing the following loss function that consists of the the Evidence Lower BOund (ELBO) (Kingma & Welling, 2014) unrolled through $N$ iterations:

$$L(\theta, \phi, (\boldsymbol{\mu}_{1,k}, \boldsymbol{\sigma}_{1,k}); \mathbf{x}) = \sum_{m=1}^M \frac{m}{M} \left[ -\log \sum_{k=1}^K \mathbf{m}_{m,k} p_\theta(\mathbf{x}|\mathbf{z}_{m,k}) + D_{\mathrm{KL}} \left( \prod_{k=1}^K q_\phi(\mathbf{z}_{m,k}|\mathbf{x}) \| p(\mathbf{z}) \right) \right] \tag{24}$$

where $p_\theta(\mathbf{x}|\mathbf{z}_{m,k})$ is the decoder log-likelihood weighted by the mask $\mathbf{m}_k$ and $D_{\mathrm{KL}}$ is the Kullback-Leibler divergence between the unit Gaussian prior $p(\mathbf{z}) = \mathcal{N}(\mathbf{0}, I)$ and the latent posterior distribution $q(\mathbf{z}_{m,k}|\mathbf{x})$ factorized across slots.

## C.5 OBJECT-CENTRIC PERCEPTION, PREDICTION, AND PLANNING (OP3)

Object-centric Perception, Prediction, and Planning (OP3) (Veerapaneni et al., 2019) extends IO-DINE to work on videos and in a reinforcement learning (RL) setting. It uses the above described IODINE as an observation model to decompose visual observations into objects and represent them independently. These representations are subsequently processed by a dynamics model that models the individual dynamics of the objects, the pairwise interaction between the objects, as well as the action's effect on the object's dynamics, predicting the next frame in latent space (Fig. C.3). By modeling the action's influence on individual objects, OP3 can be applied to RL tasks.

OP3 performs $M$ refinement steps after each dynamics step.

**Refinement network.** The refinement steps proceed as in the description for IODINE in Section C.4. The input image $\mathbf{x}_t \in [0, 1]^{H \times W \times 3}$, which is the frame from a video at time $t$, is processed by the refinement network $f_\phi$ conditioned on a sample from the current posterior estimate $\mathbf{z}_{t,m,k} \in \mathbb{R}^L$. The refinement network outputs an update of the posterior parameters $(\boldsymbol{\mu}_{t,m,k}, \boldsymbol{\sigma}_{t,m,k})$ (see Eq. (23)). The posterior parameters $(\mu_{1,1,k}, \sigma_{1,1,k})$ are randomly initialized.

**Dynamics model.** After refinement, samples from the current posterior estimate $\mathbf{z}_{t,M,k}$ for each slot $k$ are used as input to the dynamics network. The dynamics model $d_\psi$ consists of a series of linear layers and nonlinearities parameterized by $\psi$. It models the individual dynamics of the objects per slot $k$, the pairwise interaction between all combinations of objects, aggregating them into a prediction of the posterior parameters for the next time step $t+1$ for each object $k$. The full dynamics model additionally contains an action component that models the influence of a given action on each object, which we do not use in our tracking setting. The predicted posterior parameters are then used in the next time step as initial parameters for the refinement network.

$$(\boldsymbol{\mu}_{t,1,k}, \boldsymbol{\sigma}_{t,1,k}) = d_\psi(\mathbf{z}_{t-1,M,k}, \mathbf{z}_{t-1,M,[\neq k]})) \tag{25}$$

**Training.** OP3 is trained end-to-end with the ELBO used at every refinement and dynamics step, with the loss $L(\theta, \phi; \mathbf{x})$ given by:

$$\sum_{t=1}^{T} \frac{1}{T} \sum_{m=1}^{M+1} \frac{\min(m, M)}{M} \left( -\log \sum_{k=1}^{K} \mathbf{m}_{t,m,k} p_\theta(\mathbf{x}_t | \mathbf{z}_{t,m,k}) + D_{\mathrm{KL}}(\prod_{k=1}^{K} q_\phi(\mathbf{z}_{t,m,k} | \mathbf{x}_t) \| q(\mathbf{z}_{t,1,k} | \mathbf{x}_t)) \right) \tag{26}$$

where for time step 1, $q(\mathbf{z}_{1,1,k} | \mathbf{x}_1) = \mathcal{N}(\mathbf{0}, I)$.

## C.6 SCALABLE OBJECT-ORIENTED REPRESENTATION (SCALOR)

SCALable Object-oriented Representation (SCALOR) (Jiang et al., 2020) is a spatial transformer-based model that extends SQAIR (Kosiorek et al., 2018) to scale to cluttered scenes. Similar to TBA is factors the latent representations in pose, depth and appearance per object and uses spatial transformers (Jaderberg et al., 2015) to render objects in parallel. In contrast to TBA, it can handle dynamic backgrounds by integrating a background RNN that models background transitions.

**Proposal-Rejection Module:**: SCALOR uses a proposal-rejection module $g$ to discover new objects. All frames up to the current time step $x_{1:t}$ are first encoded using a convolutional LSTM $f$. The resulting features are then aggregated with an encoding of propagated object masks and divided into $H \times W$ grid cells.

$$\mathbf{c}_t^{img} = f(\mathbf{x}_{1:t}; \psi) \tag{27}$$

$$\mathbf{c}_t^{mask} = MaskEncoder(M_t^P) \tag{28}$$

$$\mathbf{c}_t^{agg} = Concat([\mathbf{c}_t^{img}, \mathbf{c}_t^{mask}], \tag{29}$$

Per grid cell a latent variable $\mathbf{z}_{t,h,w}$ is proposed. Proposal generation is done in parallel. Each $\mathbf{z}_{t,h,w}$ consists of existence, pose, depth and appearance parameters $(\mathbf{z}_{t,h,w}^{pres}, \mathbf{z}_{t,h,w}^{pose}, \mathbf{z}_{t,h,w}^{depth}, \mathbf{z}_{t,h,w}^{what})$.

$$\mathbf{z}_{t,h,w}^{pres} \sim Bern(\cdot|g_1(\mathbf{c}_t^{agg})) \tag{30}$$

$$\mathbf{z}_{t,h,w}^{depth} \sim \mathcal{N}(\cdot|g_2(\mathbf{c}_t^{agg})) \tag{31}$$

$$\mathbf{z}_{t,h,w}^{pose} \sim \mathcal{N}(\cdot|g_3(\mathbf{c}_t^{agg})) \tag{32}$$

where $g_1, g_2$ and $g_3$ are convolutional layers.

The appearance parameters $\mathbf{z}_{t,h,w}^{what}$ are obtained by first taking a glimpse from frame $x_t$ of the area specified by $\mathbf{z}_{t,h,w}^{pose}$ via a Spatial Transformer Network (STN) (Jaderberg et al., 2015) and subsequently extracting features from it via a convolutional neural network:

$$\mathbf{c}_{t,h,w}^{att} = STN(x_t, \mathbf{z}_{t,h,w}^{pose}) \tag{33}$$

$$\mathbf{z}_{t,h,w}^{what} \sim \mathcal{N}(\cdot|GlimpseEnc(\mathbf{c}_{t,h,w}^{att})) \tag{34}$$

$$\mathbf{o}_{t,h,w}, \mathbf{m}_{t,h,w} = STN^{-1}(GlimpseDec(\mathbf{z}_{t,h,w}^{what}), \mathbf{z}_{t,h,w}^{pose}) \tag{35}$$

where $\mathbf{o}_{t,h,w}$ is the object RGB glimpse and $\mathbf{m}_{t,h,w}$ is the object mask glimpse.

In the rejection phase, objects that overlap more than a threshold $\tau$ in pixel space with a propagated object from the previous time step are rejected.

**Propagation Module:**: During propagation, for each object $k$ from the previous time step $t-1$ a feature attention map $\mathbf{a}_{t,k}$ from the encoded frame features $\mathbf{c}_t^{img}$ is extracted centered on the position of the object in the previous time step and used to update the hidden state $\mathbf{h}_{t,k}$ of the tracker RNN for object $k$.

$$\mathbf{a}_{t,k} = att(STN(\mathbf{c}_t^{img}, \mathbf{z}_{t-1,k}^{pose}) \tag{36}$$

$$\mathbf{h}_{t,k} = GRU([\mathbf{a}_{t,k}, \mathbf{z}_{t-1,k}], \mathbf{h}_{t-1,k}) \tag{37}$$

$$\mathbf{z}_{t,k} = update(\mathbf{a}_{t,k}, \mathbf{h}_{t,k}, \mathbf{z}_{t-1,k}) \tag{38}$$

where $STN$ is a spatial transformer module (Jaderberg et al., 2015). If $\mathbf{z}_{t,k}^{pres} = 1$ the latent representation $\mathbf{z}_{t,k}$ of the respective object $k$ will be propagated to the next time step.

**Background:**: The background of each frame $x_t$ is encoded using a convolutional neural network conditioned on the masks $M_t$ of the objects present at time step $t$ and decoded using a convolutional neural network.

$$(\boldsymbol{\mu}^{bg}, \boldsymbol{\sigma}^{bg}) = BgEncoder(x_t, (1 - M_t)) \tag{39}$$

$$\mathbf{z}_t^{bg} \sim \mathcal{N}(\boldsymbol{\mu}^{bg}, \boldsymbol{\sigma}^{bg}) \tag{40}$$

$$\widehat{\mathbf{x}}_t^{bg} = BgDecoder(\mathbf{z}_t^{bg}) \tag{41}$$

**Rendering:**: To obtain frame reconstructions $\widehat{\mathbf{x}}_t$ foreground object appearances and masks

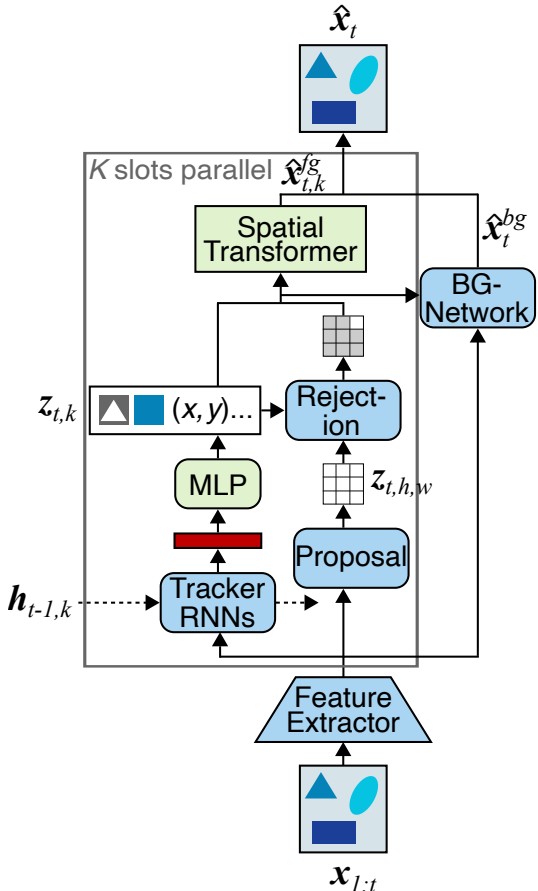

Figure C.4: **SCALOR** Feature extractor CNN $f$ followed by tracker RNNs or proposal-rejection module to compute latent $\mathbf{z}_{t,k}$. Spatial Transformer in addition to background module renders reconstruction.

are scaled and shifted using via a Spatial Transformer Network (STN):

$$\widehat{\mathbf{x}}_{t,k}^{fg} = STN^{-1}(\mathbf{o}_{t,k}, \mathbf{z}_{t,k}^{pose}) \tag{42}$$

$$\boldsymbol{\gamma}_{t,k} = STN^{-1}(\mathbf{m}_{t,k} \cdot \mathbf{z}_{t,k}^{pres}\sigma(-\mathbf{z}_{t,k}^{depth}), \mathbf{z}_{t,k}^{pose}) \tag{43}$$

$$\widehat{\mathbf{x}}_t^{fg} = \sum_K \widehat{\mathbf{x}}_{t,k}^{fg}\boldsymbol{\gamma}_{t,k} \tag{44}$$

Subsequently, foreground objects and background reconstruction are combined as follows to obtain the final reconstruction:

$$\widehat{\mathbf{x}}_t = \widehat{\mathbf{x}}_t^{fg} + (1 - M_t) \odot \widehat{\mathbf{x}}_t^{bg} \tag{45}$$

**Training::** SCALOR is trained on frame reconstruction using the evidence lower bound (ELBO):

$$\sum_{t=1}^{T} -\log p_\theta(\mathbf{x}_t|\mathbf{z}_t) + D_{\mathrm{KL}}(q_\phi(\mathbf{z}_t|\mathbf{z}_{<t}, \mathbf{x}_{\leq t})\|q(\mathbf{z}_t|\mathbf{z}_{<t})) \tag{46}$$

## D  MODEL IMPLEMENTATION DETAILS

### D.1  VIDEO MONET

**VAE:** Following (Burgess et al., 2019), the VAE encoder is a CNN with 3x3 kernels, stride 2, and ReLU activations (Table D.1). It receives the input image and mask from the attention network as input and outputs $(\mu, \log \sigma)$ of a 16-dimensional Gaussian latent posterior. The GRU has 128 latent dimensions and one hidden state per slot followed by a linear layer with 32 output dimensions. The VAE decoder is a Broadcast decoder as published by (Watters et al., 2019b) with no padding, 3x3 kernels, stride 1 and ReLU activations (Table D.2). The output distribution is an independent pixel-wise Gaussian with a fixed scale of $\sigma = 0.09$ for the background slot and $\sigma = 0.11$ for the foreground slots.

**Attention Network:** The attention network is a U-Net (Ronneberger et al., 2015) and follows the architecture proposed by (Burgess et al., 2019). The down and up-sampling components consist each of five blocks with 3x3 kernels, 32 channels, instance normalisation, ReLU activations and down- or up-sampling by a factor of two. The convolutional layers are bias-free and use stride 1 and padding 1. A three-layer MLP with hidden layers of size 128 connect the down- and the up-sampling part of the U-Net.

**Training:** MONET and VIMON are implemented in PyTorch (Paszke et al., 2019) and trained with the Adam optimizer (Kingma & Ba, 2015) with a batch size of 64 for MONET and 32 for VIMON, using an initial learning rate of 0.0001. Reconstruction performance is evaluated after each epoch on the validation set and the learning rate is decreased by a factor of 3 after the validation loss hasn't improved in 25 consecutive epochs for MONET and 100 epochs for VIMON, respectively. MONET and VIMON are trained for 600 and 1000 epochs, respectively. The checkpoint with the lowest reconstruction error is selected for the final MOT evaluation. MONET is trained with $\beta = 0.5$ and $\gamma = 1$ and VIMON is trained with $\beta = 1$ and $\gamma = 2$. $K = 5$ for SpMOt, $K = 6$ for VMDS and $K = 8$ for VOR. Due to the increased slot number for VOR, batch size for VIMON had to be decreased to 24 to fit into the GPU memory. Respectively, the initial learning rate is set to 0.000075 for VIMON on VOR. We initialize the attention network and the VAE in VIMON with the pre-trained weights from MONET to facilitate learning and speed up the training. Note that for all evaluations, the reconstructed masks $\widehat{\mathbf{m}}$ from the VAE were used.

**Sprites-MOT Initialization:** When training MONET and Video MONET on Sprites-MOT from scratch, MONET struggles to learn the extreme color values of the objects that Sprites-MOT features. Instead it completely focuses on learning the shapes. To circumvent that, we initialized the weights of the models with MONET weights that were trained for 100 epochs on Multi-dSprites.

Table D.1: Architecture of VIMON **VAE Encoder**.

| Type | Size/Ch. | Act. Func. | Comment |
|------|----------|------------|---------|
| Input | 4 | | RGB + Mask |
| Conv 3x3 | 32 | ReLU | |
| Conv 3x3 | 32 | ReLU | |
| Conv 3x3 | 64 | ReLU | |
| Conv 3x3 | 64 | ReLU | |
| MLP | 256 | ReLU | |
| MLP | 32 | Linear | |

Table D.2: Architecture of VIMON **VAE Decoder**.

| Type | Size/Ch. | Act. Func. | Comment |
|------|----------|------------|---------|
| Input | 16 | | |
| Broadcast | 18 | | + coordinates |
| Conv 3x3 | 32 | ReLU | |
| Conv 3x3 | 32 | ReLU | |
| Conv 3x3 | 32 | ReLU | |
| Conv 3x3 | 32 | ReLU | |
| Conv 1x1 | 4 | Linear | RGB + Mask |

### D.2 TRACKING BY ANIMATION

**Preprocessing:** TBA expects its input frames to contain only foreground objects. In (He et al., 2019), the authors use Independent Multimodal Background Subtraction (IMBS) (Bloisi & Iocchi, 2012) to remove the background from datasets consisting of natural videos with static backgrounds. Background subtraction algorithms maintain a spatio-temporal window around each pixel in the sequence, and remove the dominant mode based on a histogram of color values. Since the default implementation of IMBS has several hand-tuned thresholds corresponding to natural videos (e.g., for shadow suppression), it cannot be directly applied to synthetic datasets like VMDS without significant hyper-parameter tuning. We instead re-generate all of the VMDS datasets with identical objects and motion but a black background for our experiments with TBA, to mimic a well-tuned background subtraction algorithm.

**Architecture:** For SpMOT, we follow the same architecture as in (He et al., 2019), while we increase the number of slots from $K = 4$ to $K = 5$ and number of layers from $O = 3$ to $O = 4$ for VMDS. Since TBA does not model the background, this makes the number of foreground slots equal to the other models in our study.

Further, we increase the size prior parameters $U \times V$ used for the shape and appearance templates from $21 \times 21$ which is used for SpMOT, to $64 \times 64$ for VMDS, which we empirically found gave the best validation loss among $48 \times 48$, $56 \times 56$, $64 \times 64$ and $72 \times 72$. All other architectural choices are kept fixed for both datasets, and follow (He et al., 2019). Note that due to this, we trained the TBA models at its default resolution of $128 \times 128$ unlike the $64 \times 64$ resolution used by MONET and OP3.

**Training and Evaluation:** We train for 1000 epochs using the same training schedule as in (He et al., 2019). The checkpoint with the lowest validation loss is selected for the final MOT evaluation. Further, we observed that the discrete nature of the shape code used in TBA's mid-level representation leads to salt-and-pepper noise in the reconstructed masks. We therefore use a $2 \times 2$ minimum pooling operation on the final output masks to remove isolated, single pixel foreground predictions and generate $64 \times 64$ resolution outputs, similar to MONET and OP3 before evaluation.

**Deviation of SpMOT results compared to original publication:** Our results were generated with 100k training frames, while the original TBA paper (He et al., 2019) uses 2M training frames for the simple SpMOT task. Further, we report the mean of three training runs, while the original paper reports one run (presumably the best). Our best run achieves MOTA of 90.5 (Table E.1). Third, we evaluate using intersection over union (IoU) of segmentation masks instead of bounding boxes.

### D.3 OP3

**Training:** The OP3 loss is a weighted sum over all refinement and dynamics steps (Eq. (26)). For our evaluation on multi-object tracking, we weight all time steps equally. In contrast to the original training loss, in which the weight value is linearly increased indiscriminately, thus weighting later predictions more highly, we perform the linear increase only for the refinement steps between dynamics steps, thus weighting all predictions equally.

OP3, as published by (Veerapaneni et al., 2019), uses curriculum learning. For the first 100 epochs, $M$ refinement steps are taken, followed by a single dynamics step, with a final refinement step afterwards. Starting after 100 epochs, the number of dynamics steps is incremented by 1 every 10 epochs, until five dynamics steps are reached. Thus, only 5 frames of the sequence are used during training at maximum.

We chose to use an alternating schedule for training, where after each dynamics step, $M = 2$ refinement steps are taken, and this is continued for the entire sequence. Thus, the entire available sequence is used, and error is not propagated needlessly, since the model is enabled to refine previous predictions on the reconstruction before predicting again. Note that this is the schedule OP3 uses by default at test-time, when it is used for model predictive control. Note that we still use 4 refinement steps on the initial observation to update the randomly initialized posterior parameters, as in the released implementation. We split all 10-step sequences into 5-step sequences to avoid premature divergence.

We train OP3 with a batch size of 16 for 300 epochs using an learning rate of 0.0003 for VMDS and VOR and 0.0001 for SpMOT. $K = 5$ for SpMOT, $K = 6$ for VMDS and $K = 8$ for VOR are used. Larger learning rates for SpMOT led to premature divergence. Note OP3 by default uses a batch size of 80 with the default learning rate of 0.0003, this led to suboptimal performance in our experiments. Finally, training OP3 is very unstable, leading to eventual divergence in almost all experiments that have been performed for this study.

The checkpoint prior to divergence with the lowest KL loss is selected for the final MOT evaluation, as the KL loss enforces consistency in the latents over the sequence. Interestingly, the checkpoint almost always corresponded to the epochs right before divergence.

### D.4 SCALOR

**Architecture:** We follow the same architecture as in (Jiang et al., 2020). We use a grid of $4 \times 4$ for object discovery with a maximum number of objects of 10. The standard deviation of the image distribution is set to 0.1. Size anchor and variance are set to 0.2 and 0.1, respectively.

For SpMOT, background modeling is disabled and the dimensionality of the latent object appearance is set to 8.

For VMDS, the dimensionality of background is set to 3 and the dimensionality of the latent object appearance is set to 16. For object discovery, a grid of $3 \times 3$ cells with a maximum number of objects of 8 is used.

For VOR, the dimensionality of background is set to 8 and the dimensionality of the latent object appearance is set to 16.

**Hyperparameter tuning:** For VMDS, we run hyperparameter search over number of grid cells $\{3 \times 3, 4 \times 4\}$, background dimension $\{1, 3, 5\}$, maximum number of objects $\{5, 8, 10\}$ (dependent on number of grid cells), size anchor $\{0.2, 0.25, 0.3, 0.4\}$, $z^{\text{what}}$ dimenisonality $\{8, 16, 24\}$ and end value of tau $\{0.3, 0.5\}$.

For SpMOT, we run hyperparameter search over maximum number of objects $\{4, 10\}$, size anchor $\{0.1, 0.2, 0.3\}$, $z^{\text{what}}$ dimensionality $\{8, 16\}$ and whether to model background (with background dimensionality 1) or not.

For VOR, we run hyperparameter search over size anchor $\{0.2, 0.3\}$ and background dimensionality $\{8, 12\}$.

We picked best hyper parameters according to the validation loss.

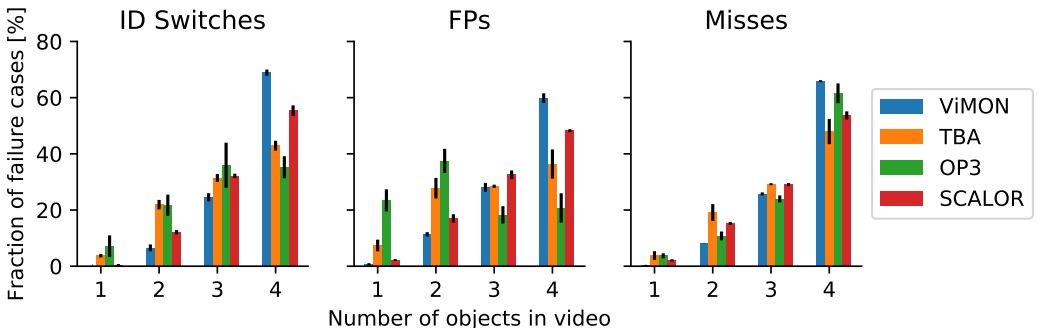

Figure E.1: Distribution of failure cases dependent on number of objects in VMDS videos split by failure class. Mean of three training runs. Error bars: SD.

**Training:** We train SCALOR with a batch size of 16 for 300 epochs using a learning rate of 0.0001 for SpMOT and VOR and for 400 epochs for VMDS. For the final MOT evaluation, the checkpoint with the lowest loss on the validation set is chosen.

# E  ADDITIONAL RESULTS

Table E.1 lists the individual results for the three training runs with different random seeds per model and dataset. The results of VIMON and SCALOR are coherent between the three runs with different random seed, while TBA has one run on SpMOT with significantly lower performance than the other two and shows variation in the three training runs on VMDS. OP3 exhibits one training run on SpMOT with lower performance than the other two.

Fig. E.1 shows the fraction of failure cases dependent on the number of objects present in the video for the three different failure cases separately; ID switches, FPs and misses. For VIMON, TBA and SCALOR, the number of failures increase with the number of objects present regardless of the type of failure. In contrast, OP3 shows this pattern for ID switches and misses, while it accumulates a higher number of false positives (FPs) in videos with fewer (only one or two) objects.

Fig. E.2 shows a comparison between MONET and VIMON on VMDS. MONET correctly finds and segments objects, but it does not assign them to consistent slots over time, while VIMON maintains a consistent slot assignment throughout the video.

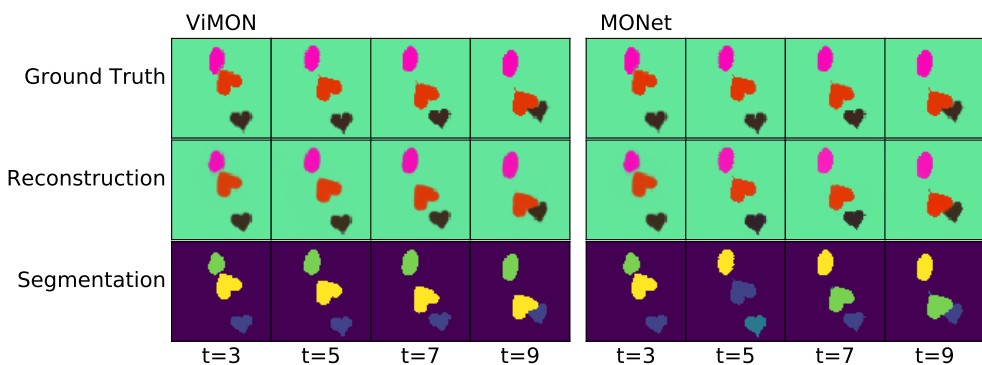

Figure E.2: Comparison of MONET and VIMON on VMDS. Example sequence of dataset shown with corresponding outputs of the model. Reconstruction shows sum of components from all slots, weighted by the attention masks. Color-coded segmentation maps in third row signify slot-assignment. Note how the object-slot assignment changes for consecutive frames (3rd row) for MONET, while VIMON maintains a consistent slot assignment throughout the video.

Fig. E.4 shows failures cases of OP3 on VOR.

Table E.2 and Table E.3 list the results for the four models, VIMON, TBA, OP3 and SCALOR, on the VMDS challenge sets and out-of-distribution (o.o.d.) sets respectively. Results are shown as the mean and standard deviation of three training runs with different random seed per model.

Table E.1: Analysis of SOTA object-centric representation learning models for MOT. Results for three runs with different random training seeds.

| Model | Run | MOTA ↑ | MOTP ↑ | MD ↑ | MT ↑ | Match ↑ | Miss ↓ | ID S. ↓ | FPs ↓ | MSE ↓ |
|-------|-----|--------|--------|------|------|---------|--------|---------|-------|-------|
| | | | | | **SpMOT** | | | | | |
| MONET | 1 | 70.0 | 90.6 | 92.8 | 49.4 | 74.7 | 4.1 | 21.2 | 4.7 | 10.4 |
| | 2 | 69.4 | 90.0 | 92.7 | 48.1 | 74.2 | 4.1 | 21.6 | 4.8 | 13.4 |
| | 3 | 71.3 | 88.1 | 91.6 | 53.8 | 77.1 | 4.9 | 18.0 | 5.8 | 15.2 |
| VIMON | 1 | 92.7 | 92.0 | 87.5 | 87.0 | 94.9 | 4.9 | 0.2 | 2.2 | 10.5 |
| | 2 | 92.8 | 92.0 | 86.9 | 86.3 | 94.8 | 5.0 | 0.2 | 2.0 | 11.8 |
| | 3 | 93.2 | 91.6 | 88.8 | 88.3 | 95.2 | 4.6 | 0.2 | 2.0 | 10.9 |
| TBA | 1 | 90.5 | 71.4 | 90.2 | 89.8 | 94.4 | 5.3 | 0.3 | 3.9 | 10.3 |
| | 2 | 58.4 | 70.7 | 69.6 | 60.8 | 75.0 | 18.1 | 6.9 | 16.6 | 14.6 |
| | 3 | 90.1 | 71.5 | 90.3 | 89.4 | 94.0 | 5.5 | 0.5 | 3.9 | 10.9 |
| OP3 | 1 | 92.4 | 80.0 | 94.5 | 93.7 | 97.3 | 2.4 | 0.4 | 4.8 | 4.3 |
| | 2 | 81.9 | 74.9 | 86.9 | 86.5 | 92.8 | 6.8 | 0.3 | 10.9 | 30.1 |
| | 3 | 92.9 | 80.1 | 95.9 | 95.2 | 97.6 | 2.0 | 0.4 | 4.7 | 5.6 |
| SCALOR | 1 | 94.4 | 80.1 | 96.5 | 92.3 | 95.4 | 2.4 | 2.2 | 1.0 | 3.3 |
| | 2 | 94.7 | 80.2 | 96.4 | 93.1 | 95.8 | 2.4 | 1.8 | 1.1 | 3.4 |
| | 3 | 95.5 | 80.2 | 96.3 | 94.0 | 96.4 | 2.4 | 1.2 | 0.9 | 3.6 |
| | | | | | **VOR** | | | | | |
| MONET | 1 | 28.0 | 81.3 | 73.8 | 26.7 | 57.4 | 18.0 | 24.6 | 29.4 | 14.1 |
| | 2 | 44.5 | 82.4 | 78.2 | 45.4 | 68.7 | 15.0 | 16.3 | 24.2 | 11.8 |
| | 3 | 38.5 | 81.6 | 78.7 | 39.8 | 67.0 | 14.4 | 18.5 | 28.5 | 10.8 |
| VIMON | 1 | 89.0 | 88.9 | 90.2 | 89.8 | 92.9 | 6.8 | 0.3 | 3.9 | 7.1 |
| | 2 | 89.0 | 89.8 | 89.9 | 89.6 | 93.0 | 6.8 | 0.2 | 4.0 | 6.2 |
| | 3 | 89.0 | 89.9 | 91.0 | 90.6 | 93.8 | 6.0 | 0.2 | 4.8 | 5.9 |
| OP3 | 1 | 64.8 | 89.5 | 87.2 | 85.1 | 90.3 | 8.8 | 0.9 | 25.5 | 3.1 |
| | 2 | 66.2 | 88.1 | 88.6 | 85.1 | 90.7 | 7.9 | 1.4 | 24.5 | 2.9 |
| | 3 | 65.3 | 89.3 | 88.2 | 86.1 | 91.1 | 8.0 | 0.9 | 25.8 | 3.0 |
| SCALOR | 1 | 74.1 | 85.8 | 75.6 | 75.5 | 77.4 | 22.6 | 0.0 | 3.3 | 6.4 |
| | 2 | 74.6 | 86.0 | 75.9 | 75.9 | 78.1 | 21.9 | 0.1 | 3.5 | 6.4 |
| | 3 | 75.1 | 86.1 | 76.5 | 76.4 | 78.2 | 21.7 | 0.0 | 3.1 | 6.3 |
| | | | | | **VMDS** | | | | | |
| MONET | 1 | 51.7 | 79.6 | 75.1 | 36.7 | 67.6 | 12.9 | 19.5 | 15.9 | 20.8 |
| | 2 | 44.3 | 76.1 | 71.8 | 34.8 | 65.9 | 15.0 | 19.1 | 21.5 | 25.3 |
| | 3 | 52.2 | 80.2 | 75.6 | 35.5 | 66.5 | 13.0 | 20.5 | 14.2 | 20.4 |
| VIMON | 1 | 87.0 | 86.8 | 86.7 | 85.4 | 92.4 | 6.8 | 0.7 | 5.5 | 10.6 |
| | 2 | 87.1 | 86.8 | 86.1 | 85.1 | 92.3 | 7.1 | 0.6 | 5.3 | 10.8 |
| | 3 | 86.5 | 86.7 | 86.0 | 84.6 | 92.1 | 7.2 | 0.7 | 5.6 | 10.6 |
| TBA | 1 | 68.5 | 76.1 | 69.3 | 65.3 | 80.7 | 16.5 | 2.8 | 12.2 | 26.0 |
| | 2 | 38.9 | 73.8 | 55.1 | 50.5 | 70.2 | 26.6 | 3.2 | 31.3 | 30.8 |
| | 3 | 56.0 | 75.0 | 64.3 | 59.2 | 76.7 | 19.8 | 3.5 | 20.8 | 27.5 |
| OP3 | 1 | 93.1 | 94.2 | 97.2 | 96.7 | 98.0 | 1.9 | 0.2 | 4.9 | 4.0 |
| | 2 | 92.7 | 93.4 | 96.9 | 96.3 | 97.8 | 2.0 | 0.2 | 5.1 | 4.3 |
| | 3 | 89.4 | 93.3 | 96.2 | 95.8 | 97.6 | 2.2 | 0.2 | 8.3 | 4.6 |
| SCALOR | 1 | 75.7 | 88.1 | 69.4 | 68.3 | 79.8 | 19.4 | 0.8 | 4.0 | 13.9 |
| | 2 | 72.7 | 87.2 | 66.7 | 65.6 | 77.6 | 21.6 | 0.8 | 4.9 | 14.2 |
| | 3 | 73.7 | 87.6 | 67.5 | 66.2 | 77.9 | 21.2 | 0.9 | 4.2 | 14.0 |

Table E.2: Performance on **VMDS challenge sets**. Results shown as mean $\pm$ standard deviation for three runs with different random training seeds. Examples sequences for each challenge set shown below.

| Model | Occlusion | | | Same Color | | | Small Objects | | | Large Objects | | |
|---|---|---|---|---|---|---|---|---|---|---|---|---|
| | MOTA | MOTP | MT | MOTA | MOTP | MT | MOTA | MOTP | MT | MOTA | MOTP | MT |
| VɪMON | 67.1 ± 0.4 | 82.5 ± 0.0 | 63.0 ± 0.1 | 72.2 ± 0.1 | 83.6 ± 0.1 | 70.4 ± 0.3 | 86.3 ± 0.2 | 83.3 ± 0.2 | 83.4 ± 0.4 | 70.7 ± 0.5 | 85.1 ± 0.1 | 76.1 ± 0.7 |
| TBA | 37.5 ± 10.4 | 72.8 ± 0.8 | 38.3 ± 4.6 | 47.2 ± 9.4 | 73.0 ± 0.7 | 45.2 ± 3.9 | 74.3 ± 0.7 | 71.9 ± 0.4 | 65.3 ± 1.6 | 25.6 ± 15.0 | 73.4 ± 0.9 | 44.7 ± 6.7 |
| OP3 | 85.3 ± 1.0 | 91.6 ± 0.4 | 89.6 ± 0.9 | 51.5 ± 1.3 | 86.5 ± 0.3 | 66.3 ± 1.3 | 93.3 ± 1.6 | 93.0 ± 0.4 | 97.0 ± 0.2 | 83.8 ± 2.0 | 92.2 ± 0.4 | 93.5 ± 0.4 |
| SCALOR | 58.8 ± 1.0 | 86.6 ± 0.4 | 46.8 ± 1.2 | 53.7 ± 1.1 | 83.4 ± 0.3 | 46.2 ± 1.1 | 74.4 ± 0.7 | 86.1 ± 0.4 | 67.6 ± 1.3 | 66.1 ± 1.9 | 86.6 ± 0.5 | 62.4 ± 1.4 |

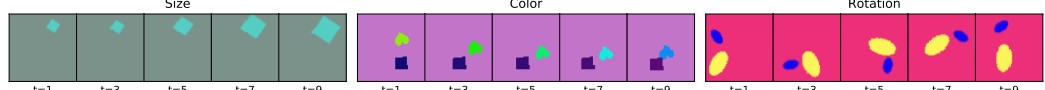

Table E.3: Performance on **VMDS OOD test sets**. Results shown as mean $\pm$ standard deviation for three runs with different random training seeds. Examples sequences for each o.o.d. set shown below.

| Model | Size | | | | Color | | | | Rotation | | | |
|---|---|---|---|---|---|---|---|---|---|---|---|---|
| | MOTA | MOTP | MD | MT | MOTA | MOTP | MD | MT | MOTA | MOTP | MD | MT |
| VɪMON | 61.4 ± 2.5 | 78.0 ± 0.3 | 71.3 ± 2.1 | 66.8 ± 1.9 | 87.4 ± 0.4 | 86.2 ± 0.2 | 86.4 ± 0.1 | 85.0 ± 0.2 | -10.4 ± 4.0 | 70.5 ± 0.4 | 39.5 ± 2.6 | 29.8 ± 1.0 |
| VɪMON* | 80.3 ± 0.9 | 82.1 ± 0.5 | 82.5 ± 0.4 | 79.8 ± 0.5 | 84.5 ± 0.6 | 84.6 ± 0.5 | 83.4 ± 0.5 | 81.8 ± 0.3 | 78.7 ± 1.6 | 82.0 ± 0.6 | 79.2 ± 0.4 | 76.4 ± 0.6 |
| TBA | 52.3 ± 8.7 | 73.3 ± 0.7 | 59.8 ± 4.9 | 51.8 ± 4.9 | 56.1 ± 11.4 | 75.1 ± 0.9 | 63.7 ± 5.4 | 59.0 ± 5.2 | 52.4 ± 9.9 | 73.6 ± 0.8 | 59.3 ± 6.2 | 49.8 ± 5.5 |
| TBA* | 1.3 ± 7.8 | 68.4 ± 1.9 | 30.6 ± 4.5 | 24.8 ± 3.4 | -16.5 ± 8.1 | 69.6 ± 1.5 | 29.1 ± 3.8 | 25.4 ± 3.3 | -7.5 ± 7.9 | 69.4 ± 1.4 | 26.6 ± 4.0 | 20.6 ± 3.4 |
| OP3 | 87.0 ± 1.9 | 90.8 ± 0.4 | 96.4 ± 0.1 | 95.3 ± 0.1 | 90.8 ± 1.2 | 93.5 ± 0.5 | 97.3 ± 0.1 | 95.8 ± 0.1 | 54.7 ± 5.7 | 84.2 ± 0.7 | 87.1 ± 1.7 | 80.5 ± 2.5 |
| OP3* | 84.0 ± 2.8 | 91.2 ± 1.0 | 95.9 ± 0.8 | 94.5 ± 1.2 | 83.6 ± 3.7 | 91.6 ± 1.3 | 95.5 ± 0.5 | 92.9 ± 1.6 | 74.5 ± 2.2 | 89.8 ± 0.7 | 94.8 ± 0.6 | 93.3 ± 0.8 |
| SCALOR | 68.1 ± 1.7 | 84.9 ± 0.4 | 63.3 ± 1.7 | 60.0 ± 2.0 | 75.5 ± 1.1 | 89.9 ± 0.5 | 67.0 ± 1.4 | 65.7 ± 1.6 | 46.5 ± 1.8 | 82.1 ± 0.5 | 41.9 ± 1.7 | 37.1 ± 1.3 |
| SCALOR* | 67.5 ± 1.2 | 85.2 ± 0.6 | 61.2 ± 1.2 | 57.1 ± 0.7 | 73.3 ± 0.7 | 89.8 ± 0.5 | 64.8 ± 1.1 | 63.0 ± 0.9 | 61.6 ± 1.4 | 83.5 ± 0.4 | 53.4 ± 1.5 | 50.2 ± 1.1 |

\* Models trained on a dataset that featured color, size and orientation changes of objects during the sequence.

## E.1 OUT-OF-DISTRIBUTION TEST SETS

To test whether the models can in principle learn additional object transformations as featured in the VMDS o.o.d. sets, we additionally train the models on a new training set that includes size and color changes as well as rotation of objects. VɪMON, OP3 and SCALOR are able to learn additional property changes of the objects when they are part of the training data while TBA fails to learn tracking on this more challenging dataset (Fig. E.3; for absolute values Table E.3).

## E.2 STABILITY OF TRAINING AND RUNTIME

To assess runtime in a fair way despite the models being trained on different hardware, we report the training progress of all models after one hour of training on a single GPU (Table E.4). In addition, we quantify inference time on the full VMDS test set using a batch size of one.

## E.3 VɪMON ABLATIONS

Removing the GRU or the mask conditioning of the attention network reduces tracking performance (MOTA on VMDS from 86.8% to 70.6% and 81.4%, respectively; Table E.5)

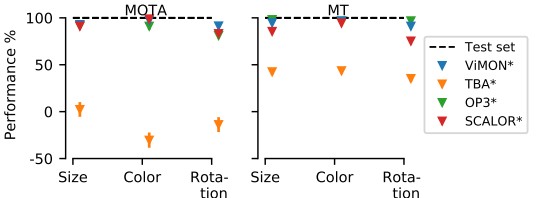

Figure E.3: Performance on out-of-distribution sets relative to VMDS test set (100%). * indicates that models were trained on a dataset that included color, size and orientation changes of objects.

## F SUPPLEMENTARY FIGURES

See figures F.1 – F.8 for additional, randomly picked examples of reconstruction and segmentation for VɪMON, TBA, OP3 and SCALOR on the three datasets (VMDS, SpMOT and VOR).

Table E.4: Runtime analysis (using a single RTX 2080 Ti GPU). Training: models trained on VMDS for one hour. Inference: models evaluated on VMDS test set with batch size=1 (10 frames).

| Model | Resolution | No. Param. | Training | | | | Inference | | |
| --- | --- | --- | --- | --- | --- | --- | --- | --- | --- |
| | | | Batch Size | Memory [MiB] | No. Iters | Epochs | Memory [MiB] | Avg. runtime / batch | Total runtime |
| VIMON | 64×64 | 714,900 | 18 | 10,860 | 3687 | 6.63 | 910 | 0.28 s/it | 4min 39s |
| TBA | 128×128 | 3,884,644* | 64 | 10,564 | 4421 | 28.29 | 972 | 0.24 s/it | 4min 05s |
| OP3 | 64×64 | 876,305 | 10 | 10,874 | 2204 | 2.20 | 4092 | 0.54 s/it | 9min 04s |
| SCALOR | 64×64 | 2,763,526 | 48 | 10,942 | 2547 | 12.23 | 930 | 0.29 s/it | 4min 48s |

* The TBA parameter count scales with the feature resolution, which is kept fixed using adaptive pooling. This makes the parameter count independent of input resolution.

Table E.5: Ablation experiments for VIMON on VMDS.

| Model | MOTA ↑ | MOTP ↑ | MD ↑ | MT ↑ | Match ↑ | Miss ↓ | ID S. ↓ | FPs ↓ | MSE ↓ |
| --- | --- | --- | --- | --- | --- | --- | --- | --- | --- |
| VIMON W/O MASK CONDITIONING | 70.6 | 87.8 | 75.7 | 66.0 | 81.4 | 13.4 | 5.2 | 10.8 | 16.9 |
| VIMON W/O GRU | 81.4 | 86.9 | 79.8 | 77.3 | 88.2 | 10.3 | 1.4 | 6.8 | 18.9 |

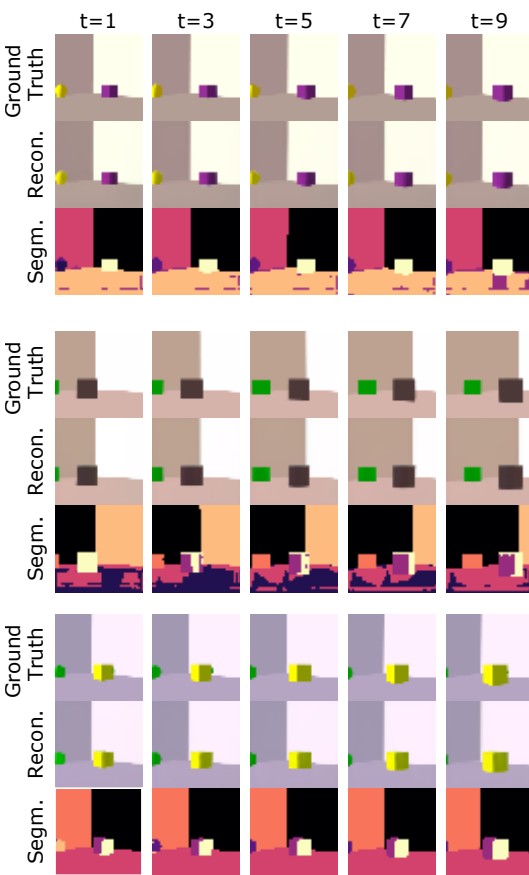

Figure E.4: Failure cases of OP3 on VOR. Example sequences of VOR test set shown with corresponding outputs of the model after final refinement step. Binarized colour-coded segmentation maps in third row signify slot-assignment.

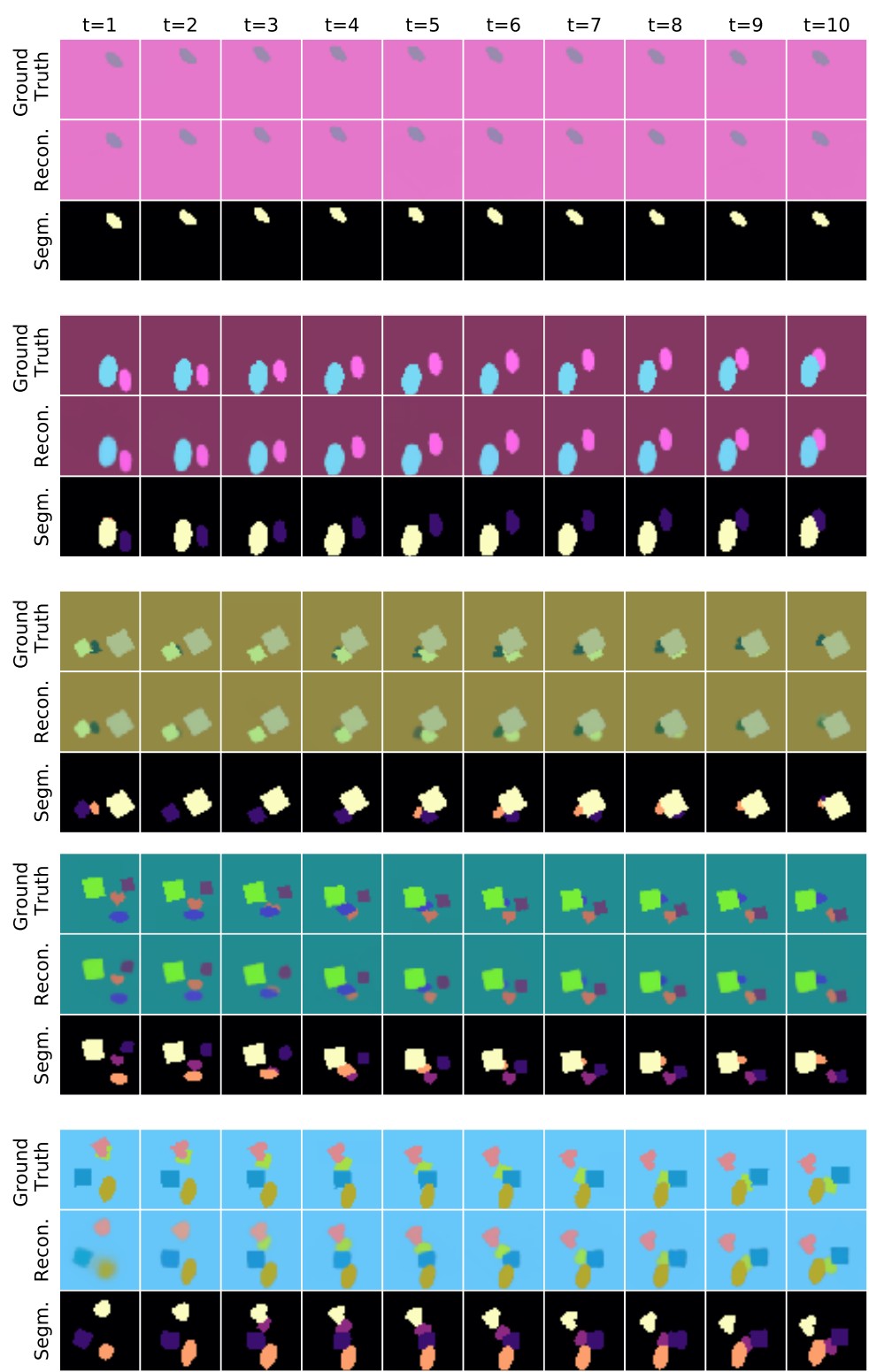

Figure F.1: Results of VIMON on VMDS. Random example sequences of VMDS test set shown with corresponding outputs of the model. Reconstruction shows sum of components from all slots, weighted by the reconstructed masks from the VAE. Binarized colour-coded segmentation maps in third row signify slot-assignment.

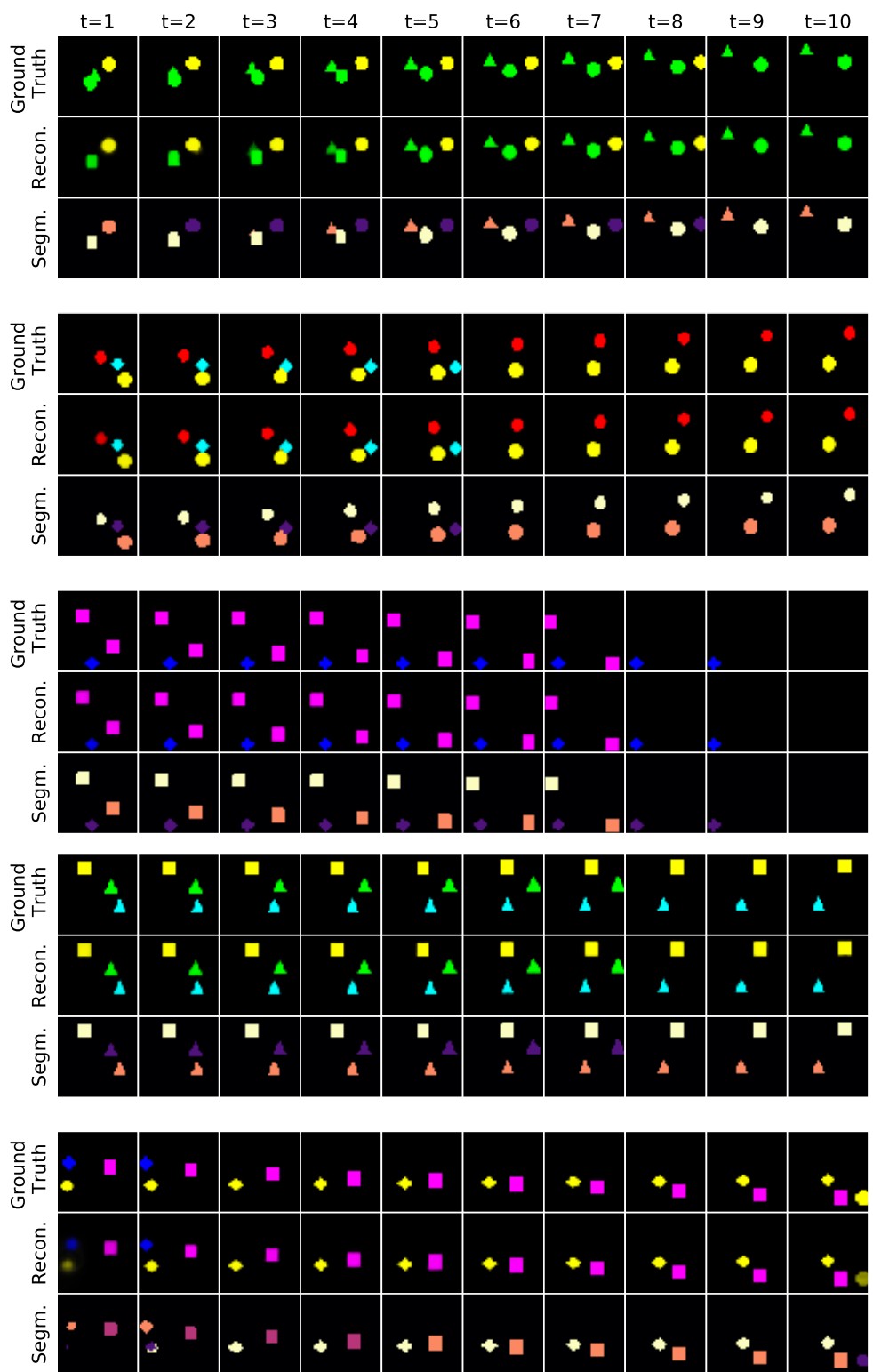

Figure F.2: Results of VIMON on SpMOT. Random example sequences of SpMOT test set shown with corresponding outputs of the model. Reconstruction shows sum of components from all slots, weighted by the reconstructed masks from the VAE. Binarized colour-coded segmentation maps in third row signify slot-assignment.

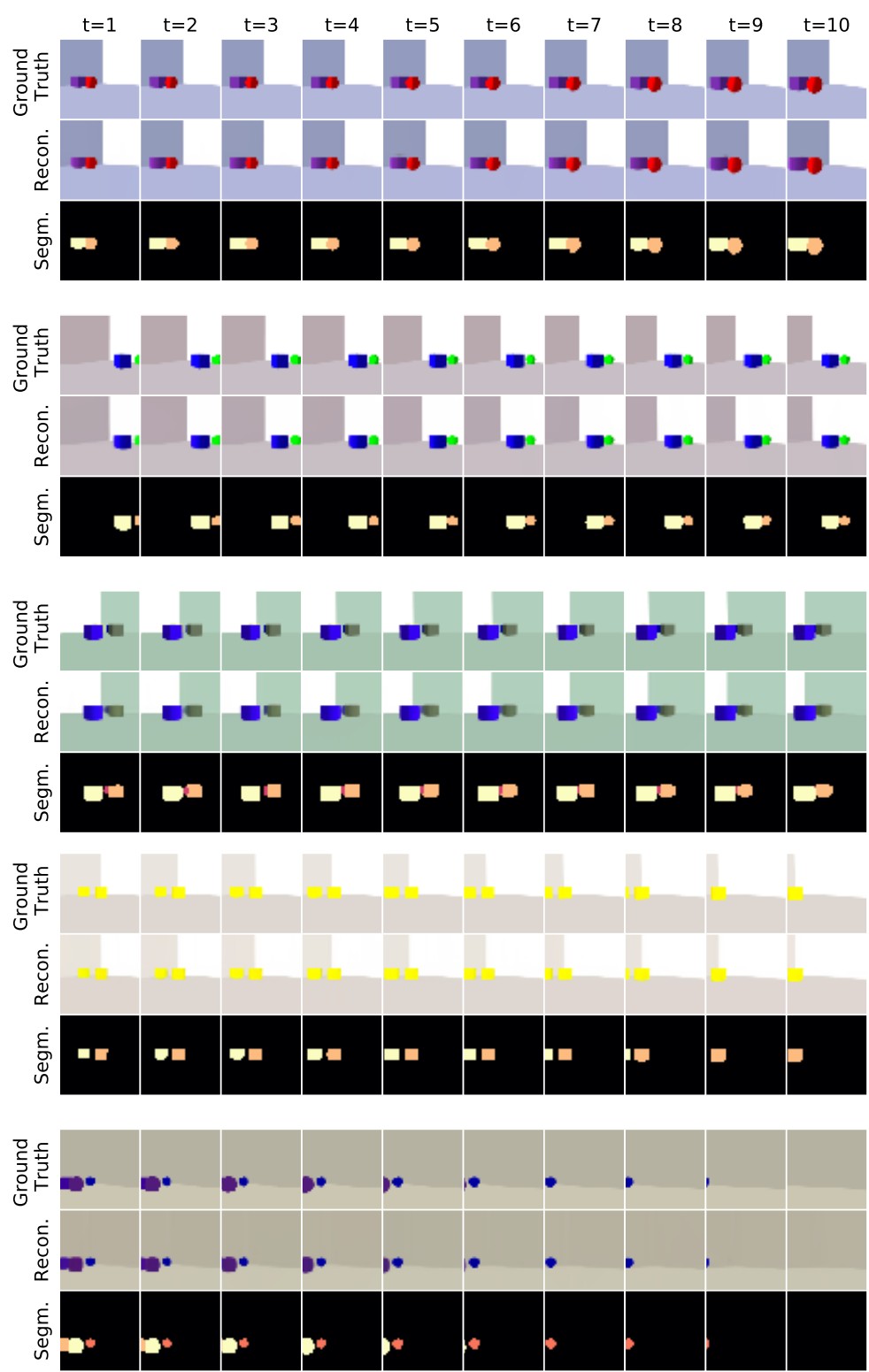

Figure F.3: Results of VɪMON on VOR. Random example sequences of VOR test set shown with corresponding outputs of the model. Reconstruction shows sum of components from all slots, weighted by the reconstructed masks from the VAE. Binarized colour-coded segmentation maps in third row signify slot-assignment.

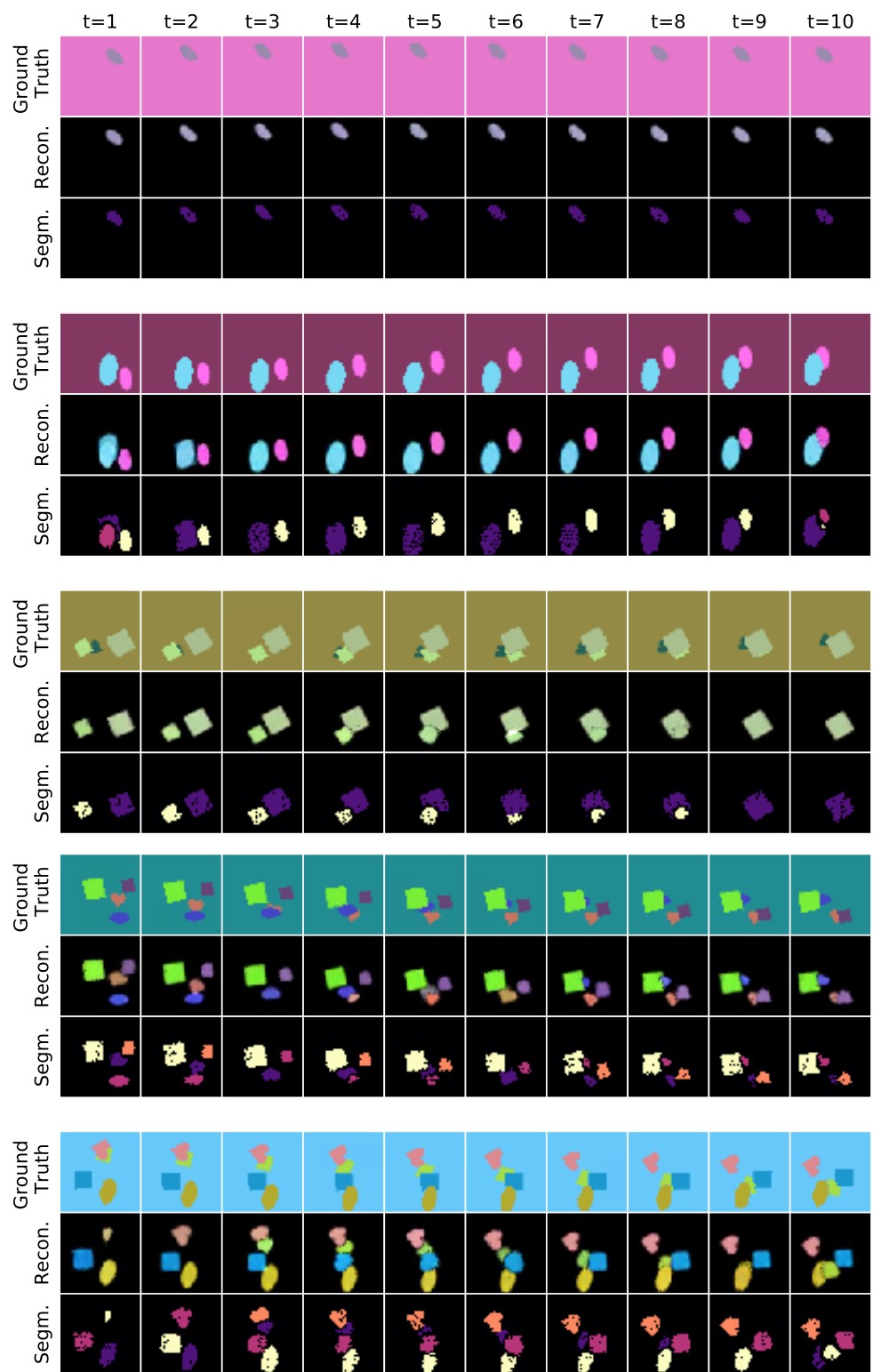

Figure F.4: Results of TBA on VMDS. Random example sequences of VMDS test set shown with corresponding outputs of the model. Binarized colour-coded segmentation maps in third row signify slot-assignment. Note that background subtraction is performed in the preprocessing of TBA, hence the black background in the reconstructions.

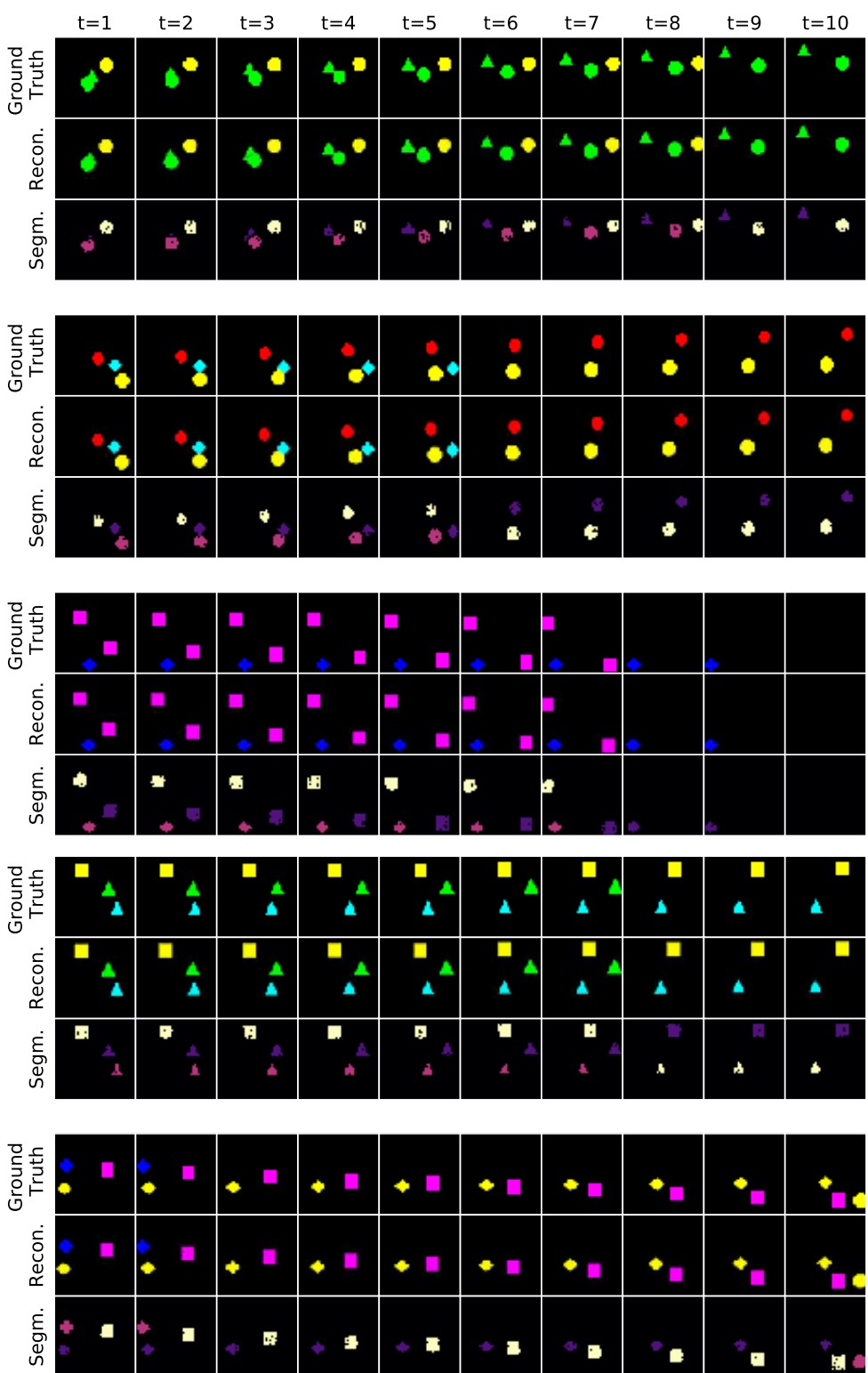

Figure F.5: Results of TBA on SpMOT. Random example sequences of SpMOT test set shown with corresponding outputs of the model. Binarized colour-coded segmentation maps in third row signify slot-assignment.

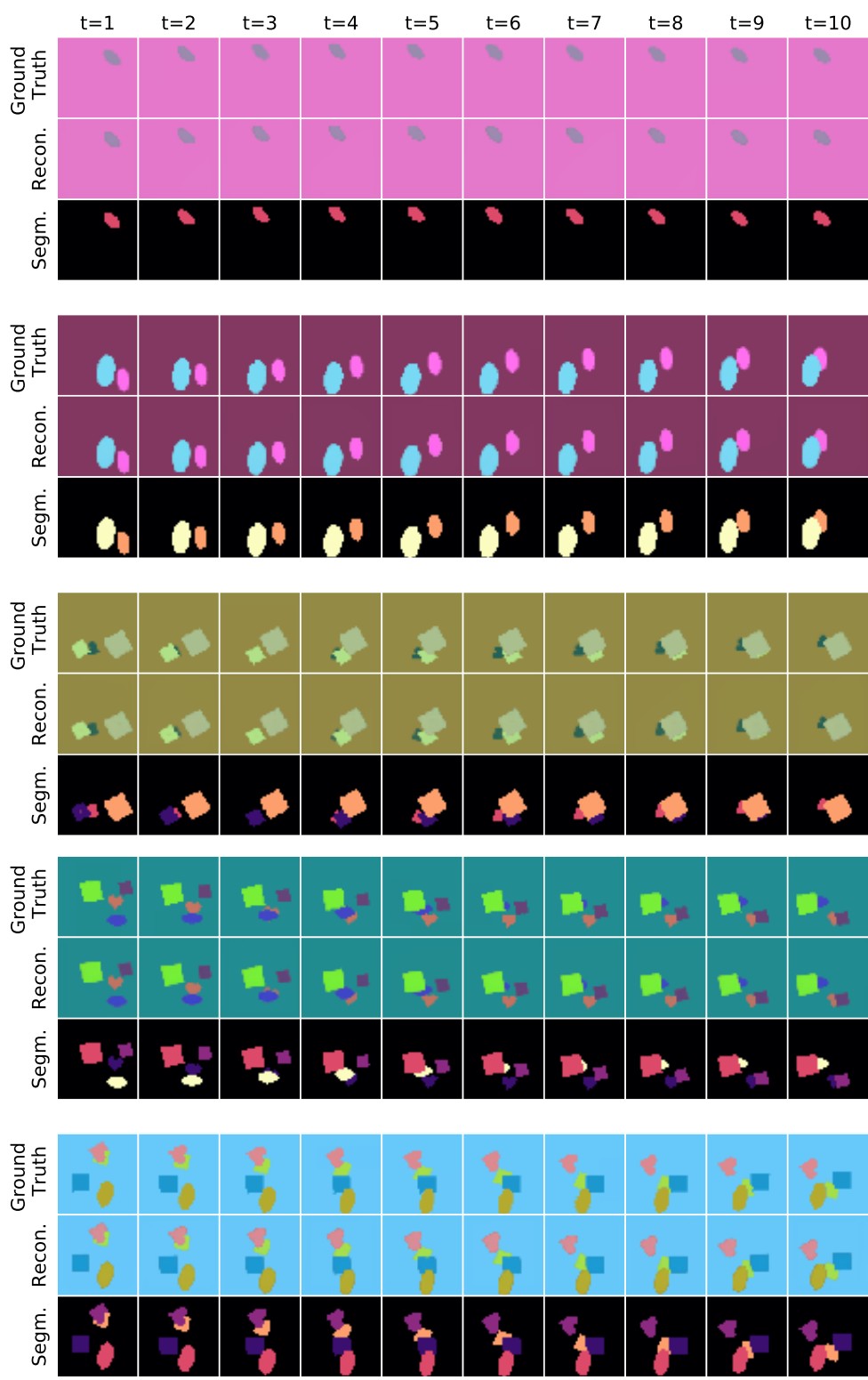

Figure F.6: Results of OP3 on VMDS. Random example sequences of VMDS test set shown with corresponding outputs of the model after final refinement step. Binarized colour-coded segmentation maps in third row signify slot-assignment.

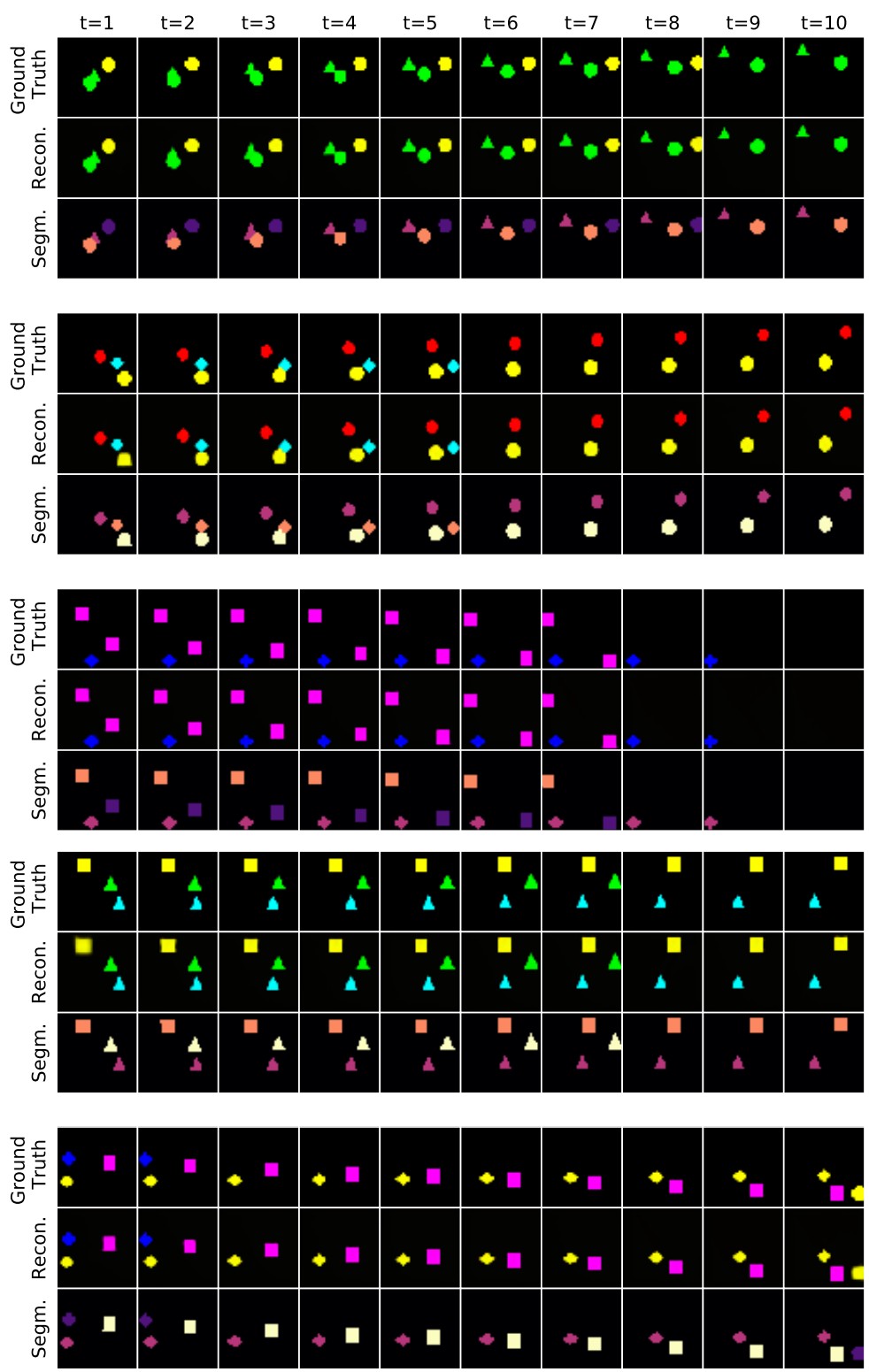

Figure F.7: Results of OP3 on SpMOT. Random example sequences of SpMOT test set shown with corresponding outputs of the model after final refinement step. Binarized colour-coded segmentation maps in third row signify slot-assignment.

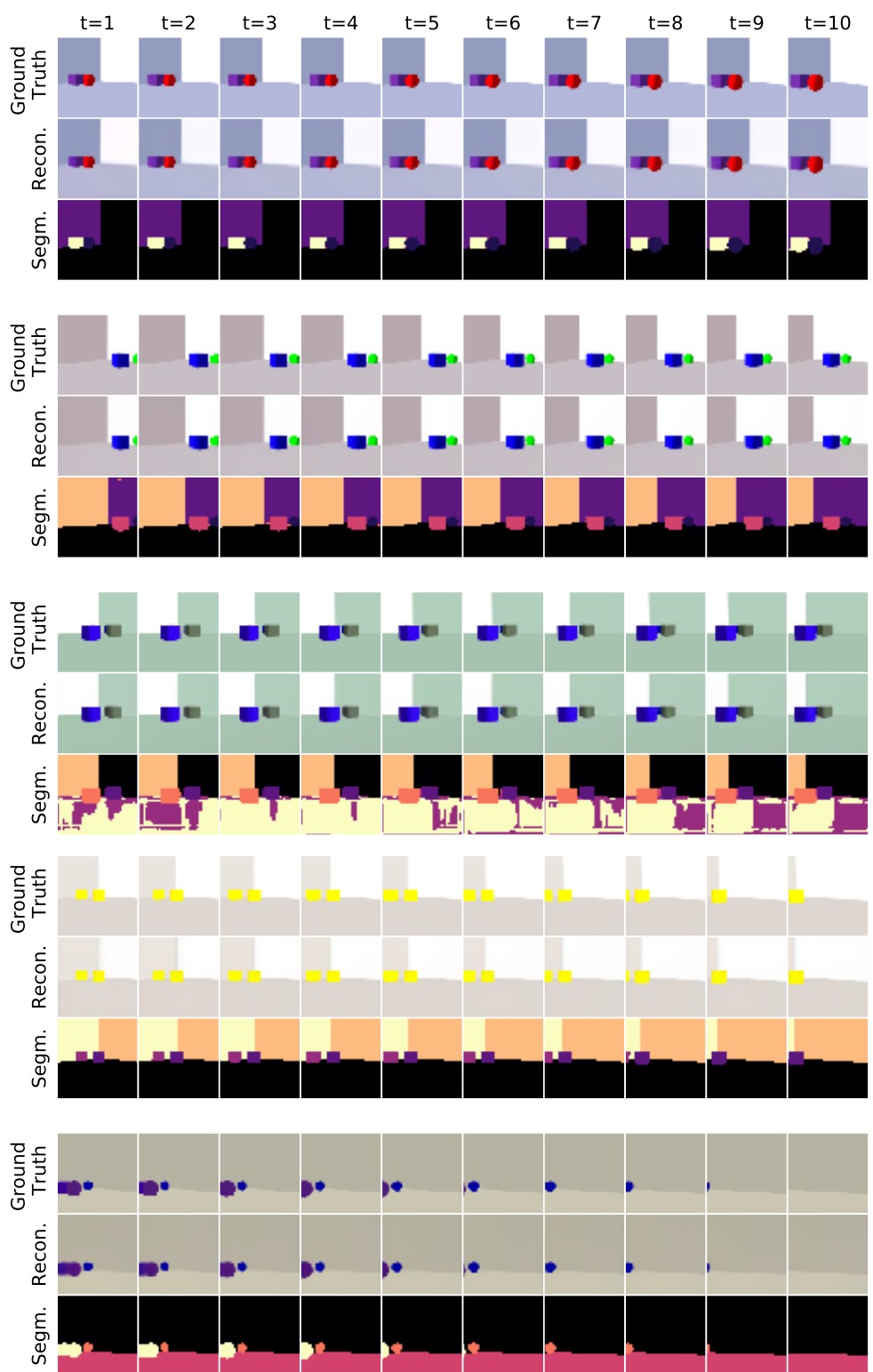

Figure F.8: Results of OP3 on VOR. Random example sequences of VOR test set shown with corresponding outputs of the model after final refinement step. Binarized colour-coded segmentation maps in third row signify slot-assignment.

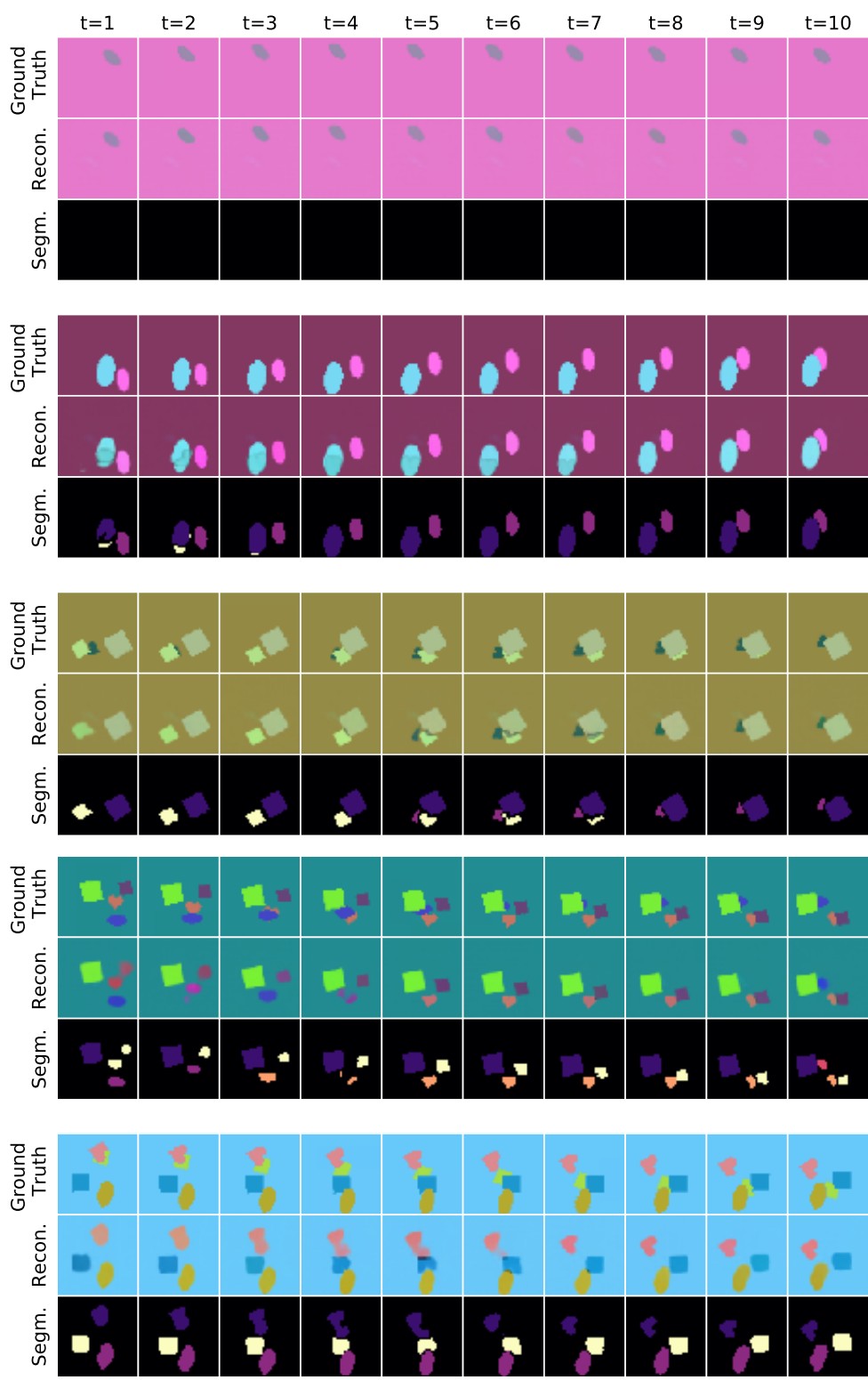

Figure F.9: Results of SCALOR on VMDS. Random example sequences of VMDS test set shown with corresponding outputs of the model. Binarized colour-coded segmentation maps in third row signify slot-assignment.

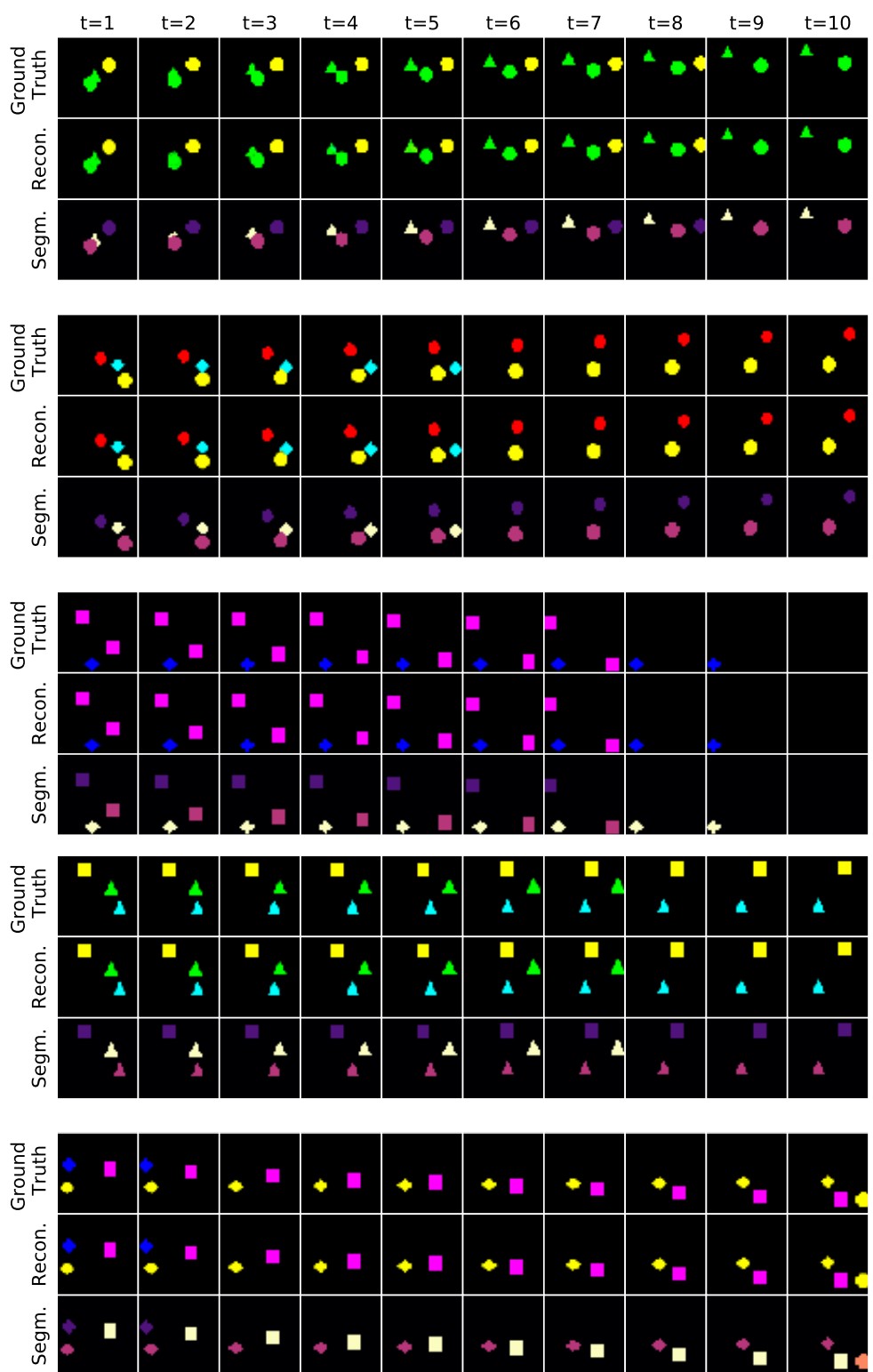

Figure F.10: Results of SCALOR on SpMOT. Random example sequences of SpMOT test set shown with corresponding outputs of the model. Binarized colour-coded segmentation maps in third row signify slot-assignment.

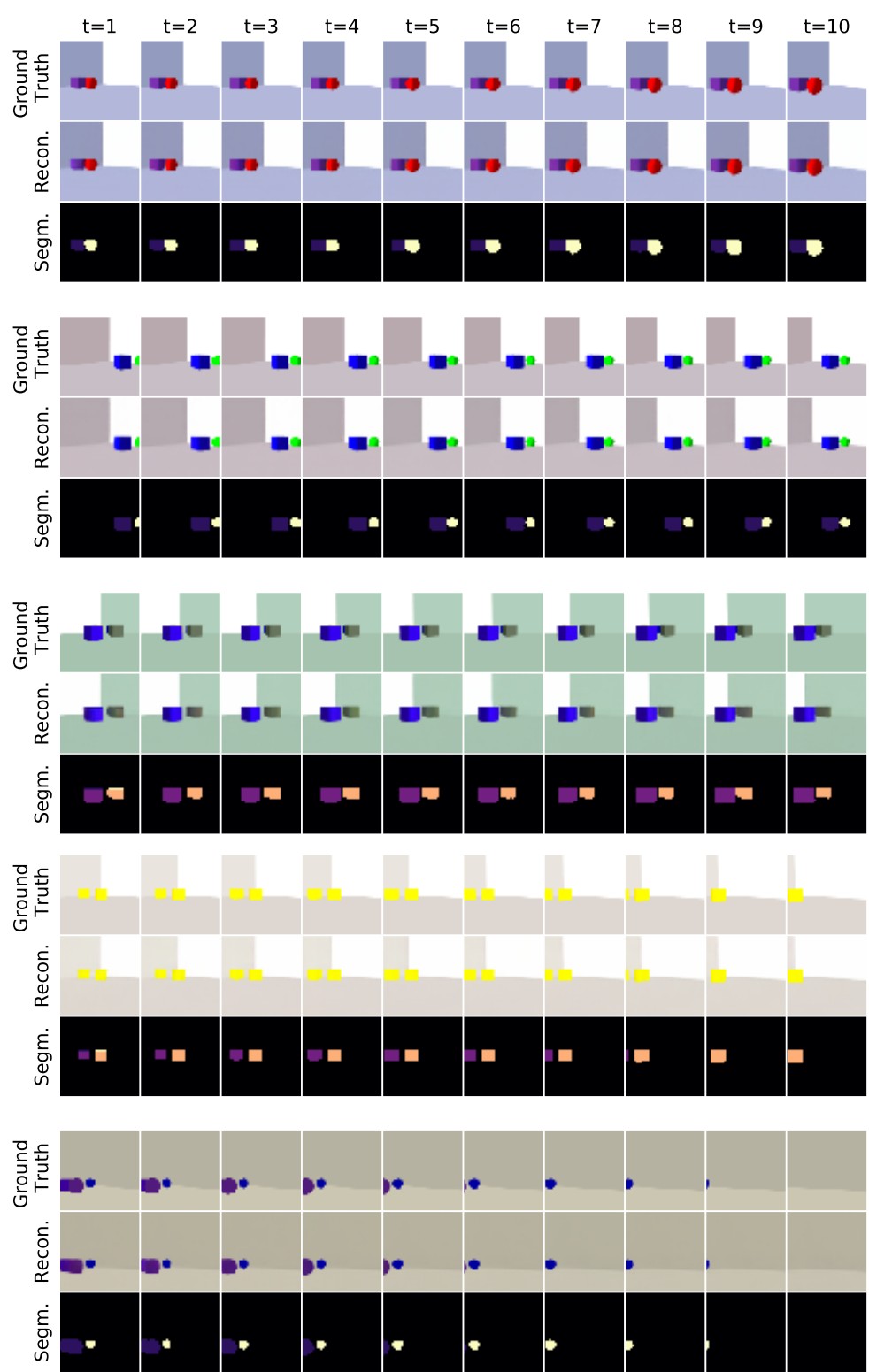

Figure F.11: Results of SCALOR on VOR. Random example sequences of VOR test set shown with corresponding outputs of the model. Binarized colour-coded segmentation maps in third row signify slot-assignment.

