# OpenReview forum: "Benchmarking Unsupervised Object Representations for Video Sequences"
_ICLR.cc/2021/Conference — Reject_

### Official Review · AnonReviewer4 · 2020-10-28
**The paper provides a benchmark for unsupervised object representations with 5 state-of-the-art methods and 3 simulation datasets.**

**Rating:** 5
**Confidence:** 2

**Review:**

Overall, this paper is interesting in setting up a benchmark for unsupervised object representations which is a very important problem in computer vision, reinforcement learning, etc. But the paper contains a very small amount of information to make use of the benchmark. Following are my comments:

1. The paper needs to clearly set the contributions. The video datasets are from self-made simulations or taken from other sources because the appendix cites many references for each part of the dataset.

2. The appendices are linked in the paper but given in the supplementary. It is good to combine both in one paper.  The appendix doesn't require information of existing methods and the current version contains of appendix contain mostly these explanations.

3. Apart from these, the paper is well written and useful in the community. But the empirical evaluation is not convincing. There could be many baseline approaches for this, for example, paper [1] and the methods it is comparing with. This is my major concern on the paper and if stated well, the recommendation can change.

[1] SPACE: Unsupervised Object-Oriented Scene Representation via Spatial Attention and Decomposition, ICLR 2020

---

> ### Author Response · Authors · 2020-11-12
> **Response to comments; feedback welcome.**
>
> Thank you for reviewing our paper and your valuable comments.
>
> Could you please clarify your sentence “the paper contains a very small amount of information to make use of the benchmark”? Does this statement refer to points 1–3 below or what kind of additional information are you missing? Note that we will make our code for models and evaluation public, as well as a leaderboard to which future methods can be added.
>
> Below we respond to your three comments. In particular for the third one, we would be interested in your feedback.
>
> 1. The SpMOT dataset is taken from He et al. (2019) and the only dataset that we did not create ourselves (it is also the dataset with the lowest visual complexity). VMDS and VOR are two datasets that we generated by taking existing image datasets and animating them to create video sequences. VMDS was generated by using the shapes from the dSprites dataset (Matthey et al., 2017). dSprites is a black-and-white image dataset with one object per image. We used that to create colored videos with multiple objects per video by sampling shapes from dSprites, randomly coloring them with RGB colors and sampling a trajectory for each object from a Gaussian process. Additionally to the training, validation and test set, we create seven challenge sets that feature different challenging tracking scenarios. VOR is a video dataset that we newly generated using the graphics engine OpenGL. Again, we used the basic 3D shapes of an earlier image dataset ObjectRoom and created animated videos by moving the camera. Please see also Appendix B1-B3 for a more detailed description of the dataset generation. We are happy to provide additional clarification if needed.
>
> 2. We are sorry if our format of submission caused any confusion. We are happy to merge the main paper and the appendix into one document.
> Regarding our description of the models in the appendix, it was meant to serve two purposes: (1) We wanted the paper to be as self-contained as possible without requiring the reader to have to go back to the original model publications. (2) We unified the mathematical notation of the methods in order to facilitate comparison between them.
>
> 3. Note that the baseline you suggested (SPACE) and methods they compare to in their paper (SPAIR, IODINE, GENESIS) are models for images, not videos. As such, these models have no mechanisms built-in for tracking objects through time, which is one of the major topics of our analysis. We used MONet merely as a simple sanity check that image-based models are indeed insufficient. It serves this purpose well since it is structurally otherwise identical to ViMON, showing that indeed temporal processing is required. Therefore, we are not sure that adding additional image-based models as baselines would be very insightful, because we expect these models to be very far from being competitive. Could you please let us know whether you still think that image-based models are important baselines and, if so, what insights you expect to gain from such baselines?

---

### Official Review · AnonReviewer2 · 2020-10-31
**A nice benchmark, but not enough**

**Rating:** 4
**Confidence:** 1

**Review:**

The paper proposes a benchmark for the evaluation of unsupervised learning of object-centric representation. The benchmark consists of three datasets, multi-object tracking metrics and of the evaluation of four methods. The proposed dataset consists of three sets of video sequences, procedurally generated, which are either generated from slight variations of existing works (Sprites-MOT) or on the basis of existing datasets (dSpirites, Video Object Room). For evaluation, authors propose to use a slight variation of the protocol of the MOT challenge for evaluation (with the addition of a Mostly Detected measure which does not penalize ID switches). As part of the  paper, they also evaluate and discuss the performances of four object-centric representation models, one of them (Video MONet) being an extension of an existing approach, proposed as part of this paper, and the remaining being state of the art approaches for the task.

**Paper Strengths**
- Although I am not familiar with unsupervised learning of object-centric representation, I like the idea of proposing a common protocol for evaluation. Potentially, this can be useful for the community and can help to create a common ground for evaluation.
- The benchmark is comprehensive, in that it contains both data and evaluation measures. The adoption of MOT metrics also appears to be a reasonable choice.
- The comparison between different models is interesting, and authors have also added a custom designed novel method (Video MONet). They investigate a set of challenging scenarios and carry out out-of-distribution tests.
- The paper is a pleasure to read, and authors have also made a good effort in writing a clear and comprehensive supplemental.

**Paper Weaknesses**
My main concern about the paper is the lack of novelty. While I realize that the objective of the paper is to create a common evaluation background, I fail to see a sufficient level of quality and of novelty. In particular:
- the dataset associated with the benchmark are mostly based on existing works - they might be appropriate for evaluating this task, but the level of contribution is a bit limited;
- on the metrics, the only contribution is to suggest using MOT metrics, which are again pre-existing;
- the experimental and the insights it gives, again, can foster the community towards better model, but it's not a sufficient contribution for ICLR in my view.

Overall, I think the paper could be a nice contribution to the literature on unsupervised learning of object-centric representation, but it lacks sufficient contribution for ICLR, in my view. I would therefore suggest to reject the paper.

---

> ### Author Response · Authors · 2020-11-20
> **Clarification of contributions**
>
> Thank you for reviewing our paper and providing critical feedback. We believe your assessment of the strengths of the paper is spot on – in particular the first point: The relatively young field of object-centric representation learning has been lacking a common framework for evaluating methods and, as a consequence, also the evaluation itself (see also AnonReviewer5). Our paper fills this gap. We believe that such an evaluation is currently one of the most critical missing pieces for the field, and at least as valuable as proposing “yet another method.”
>
> We evidently did not do a good job at describing our contributions clearly. Let us clarify.
>
> Contribution 1: Establish that there is a problem. The main contribution of our paper is showing that none of the methods work as promised in the original papers. We did not state this point that explicitly, perhaps in an attempt at being diplomatic. Specifically:
>
> - All models rely heavily on color and struggle with objects of similar color, which is a strange result since color is a noiseless, almost perfect cue in the sprites datasets. A simple edge detector followed by a non-zero threshold and some trivial morphological operations would already segment objects perfectly, yet current neural network models do not learn to disentangle similarly colored objects despite thousands of training samples (cp. Fig. 4e). This is clearly a suboptimal solution since not separating even similarly colored objects both incurs a higher reconstruction error as well as requires a more expressive latent space than properly disentangling them.
>
> - Despite explicitly encoding depth, neither TBA nor SCALOR handle occlusion more gracefully than our simpler ViMON model, showing that inferring depth is still an unsolved problem even on such simple synthetic datasets.
>
> - OP3, while being the model most robust to occlusion, suffers from unstable training and generates many false positives when there are fewer objects in the scene than the model has slots.
>
> Documenting these findings in the publication record is important, because it affects how people entering the field will choose what problems to work on, which affects how fast the community as a whole can make progress (as opposed to only insiders who may have realized some of these points independently already).
>
> Contribution 2: Establish a way to measure progress. The state of the field before our paper was that it wasn’t clear how existing methods fare relative to each other in different respects (tracking, occlusion handling, depth reasoning, segmentation, …). We provide this comparison along with a dataset (VMDS) with several challenge sets that establish quantitative metrics to measure progress in the future. This dataset (and, similarly, VOR) is loosely inspired by earlier work in the sense that it uses the same four basic sprites, but apart from that is entirely novel: we create videos with smoothly moving sprites, add color as an additional cue and create several challenge sets that systematically vary certain properties of the movies to allow us to disentangle sources of difficulty for the models.
>
> Contribution 3: Provide suggestions for how to move forward. We realized that many of the conclusions in the Discussion section were implicit. We therefore revised it to provide more explicit suggestions as pointed out by AnonReviewer1.
>
> In summary, we kindly but strongly disagree with your assessment of the level of contribution for ICLR. In our opinion, the paper fills one of the key missing pieces in the field of object-centric representation learning at the moment and therefore clearly constitutes an important contribution to a conference focused on representation learning such as ICLR.

---

### Official Review · AnonReviewer1 · 2020-11-03
**The paper provides a framework to compare five models on four synthetic datasets for object detection/segmentation and tracking .**

**Rating:** 5
**Confidence:** 3

**Review:**

The paper is actually very well written and tries to answer the question of how various unsupervised learning of object-centric representations to on controlled tasks (synthetic datasets). The positives:
1. Designs a benchmark of three datasets of varying complexity
2. Compare a single image model and four video models (total five)
3. Defined clear metrics that are around precision, detection, segmentation, tracking

The cons to the paper:
1. All of the datasets are synthetic and it would have been good to at least pick a real world dataset and confirm the conclusions stand
2. Since, the core goal of the paper is not novelty but a better understanding of various models I would have liked for the discussion to have some clear conclusions and better structure (use X in scenario Y, Model Z needs to be extended for scenarios Y etc.). I feel this section was short and mostly verbose without a clear conclusion
3. Instead of focusing on such a comprehensive set of things - 3 datasets, five models and multiple metrics it would have been better if authors did some minor extensions of the models and showcase novel directions. But, instead the paper is mostly an understand only work and i worry that the conclusions don't necessarily give clear future directions for other researchers to build on.

Given the cons and specifically on not a clear actionable suggestion on how to improve models and no analysis beyond synthetic datasets I am leaning towards a rating of below acceptance threshold.

---

> ### Author Response · Authors · 2020-11-12
> **Clarification of dataset selection**
>
> Thank you for your constructive comments. We would like to clarify and get your input on your first criticism that we use only synthetic datasets; a response to the other points will follow.
>
> - We focus on synthetic datasets, because current object-centric models are not capable of modeling the visual complexity of real-world videos. This limitation is well-known in the literature [1] and existing papers focus on synthetic datasets (or use preprocessing like background removal to essentially turn them into sprite-like datasets [2]).
> - Synthetic stimuli enable us to precisely generate challenging scenarios in a controllable manner (occlusion, role of color, …). Such analyses would not be possible with real-world datasets, even if the models were able to handle them.
>
> Having said that, it is possible that we are unaware of a suitable dataset on which current object-centric learning models succeed. If you think that is the case, which dataset would you suggest we add to our benchmark?
>
> [1] Greff et al. Multi-object representation learning with iterative variational inference. In Proc. of the International Conf. on Machine learning (ICML), 2019.
>
> [2] He et al. Tracking by animation: Unsupervised learning of multi-object attentive trackers. In Proc. IEEE Conf. on Computer Vision and Pattern Recognition (CVPR), 2019.

---

> ### Author Response · Authors · 2020-11-20
> **Response to comments 2+3**
>
> Thanks again for your critical feedback. Here we would like to address your second and third concern.
>
> We realize that many of our conclusions were only implicit in the discussion, but we missed spelling them out explicitly. Based on your feedback, we updated the discussion and formulated the implications of our findings more concretely; we also list them again below:
>
> - Occlusion handling is a key component object-centric video models need to master, but not yet sufficiently do. Moving forward, the models that don’t include a potent interaction module like OP3’s that takes pairwise interaction between objects (including occlusion) into account, could be improved by incorporating one. Preferable, though, would be making sure that the depth reasoning of the models works as intended (see next point).
> - Teaching models to make use of amodal masks in combination with proper depth reasoning would enable much more compact latents, which are necessary to make proper use of the intuitions behind object-centric learning, namely that decomposing a scene into objects is more efficient than representing multiple or all of them at once. This would help prevent multiple objects from getting segmented in one slot as well as facilitate reasoning about occlusion.
>
> - To scale to more natural data and prevent the methods from missing objects that are similar to the background, the pixel-wise reconstruction might need to be replaced, for instance by using contrastive learning [1] or perceptual loss functions [2,3].
>
> - Choosing a class of models is dependent on the dataset one wants to apply it to as well as on the computational resources at one's disposal. According to our experiments, datasets that feature a high number of objects (>10) that are well separated from each other make a method like SCALOR, which can process objects in parallel, advisable. On datasets with a lower number of objects per scene which feature heavy occlusion, methods like OP3 and ViMON will likely achieve better results, but require a high computational budget for training. Having said that, no model was able to handle all tracking scenarios in our benchmark gracefully, suggesting that in order for the field to move forward advantages of the different models need to be combined, as well as new models need to be rigorously evaluated on these challenging, systematic evaluation scenarios.
>
> Regarding your third point that we should rather do some minor extensions: Resolving the issues we demonstrate is a logical next step. However, these are difficult problems that are not resolved by minor tweaks. Moreover, to test whether an extension actually helps at resolving X, we need datasets and evaluation protocols that test for X. Furthermore, we need to know how the existing approaches perform. The value of our paper lies in establishing such datasets, protocols and performing a comprehensive evaluation that enables the broader community to work on novel approaches and make progress.
>
>
> [1] Kipf et al. Contrastive learning of structured world models. ICLR 2020.
>
> [2] Gatys et al. A neural algorithm of artistic style. arXiv.org, 1508.06576, 2015.
>
> [3] Hou et al. Deep feature consistent variational autoencoder. WACV 2017.

---

### Official Review · AnonReviewer5 · 2020-11-06
**A timely and well-executed evaluation of existing methods**

**Rating:** 7
**Confidence:** 3

**Review:**

The paper presents an empirical evaluation of a number of recent models for unsupervised object-based video modelling. Five different models are evaluated on three (partially novel) benchmarks, providing a unifying perspective on the relative performance of these models. Several common issues are identified and highlighted using challenge datasets: The reliance on color as a cue for object segmentation, occlusion, object size, and change in object appearance. The paper concludes with several ideas for alleviating these issues.

Strengths:
 1. The paper represents a much needed comparison of several related models which have previously not been evaluated on common benchmarks. Given the rapidly increasing number of competing models in this space, I believe analysis papers like this one serve an important role.
 2. The paper highlights important weaknesses of unsupervised object models, such as the overreliance on color or the difficulties with handling occlusion. While I believe that some of these weaknesses were already known to the people working with these methods, they have not always been formally documented in the respective publications. This paper rectifies this, and also provides guidance as to the relative vulnerability of the different methods.
 3. The methodology is convincing and thorough. Architecture and hyperparameter choices are clearly documented in the appendix, and the datasets have been published.

Weaknesses:
1. As an analysis paper, this publication does not provide specific new technical contributions to the issues it is evaluating.
2. The results are not entirely conclusive, in that there is no clear best model, and their relative quality varies with datasets and metrics. That said, some things can be very clearly observed, for instance the importance of object size for the performance of TBA.

Overall, the paper serves an important role in consolidating the ecosystem of unsupervised object representations. Given the increasing need for such analysis papers, and the competent execution, I recommend acceptance.

---

> ### Author Response · Authors · 2020-11-20
> **Response to comments**
>
> Thank you for your constructive feedback and stressing that our work is much needed, important and competently executed.
>
> Two quick points regarding the weaknesses you state:
>
> 1. Note that we do make a technical contribution by introducing ViMON, which is a competitive model that is simpler than many of its competitors.
>
> 2. As you correctly point out, there is no overall best model in our benchmark. Selecting an overall winner was not the intention of our benchmark. Given that object-centric models so far are only working on toy data and are not deployable in real-world scenarios, our paper is focused on basic research that aims at understanding fundamental problems. Therefore, we tried to give a nuanced assessment of the models under different circumstances to point out their respective strengths and weaknesses to guide future research. We revised our discussion to emphasize these implications more clearly.

---

### Decision · Program_Chairs · 2021-01-07
**Final Decision**

**Decision:**

Reject

**Comment:**

This paper received 4 reviews with mixed initial ratings: 7, 5, 4, 5. The main concerns of R1, R2 and R4, who gave unfavorable scores, included: lack of methodological novelty (analysis-only paper), absence of experiments on real data (3 synthetic-only benchmarks), missing baselines and an overall inconclusive discussion. At the same time R5 notes that the offered fair comparison between SOTA methods was indeed "much needed", and the paper can "serve an important role" in guiding future developments in the community. In response to that, the authors submitted a new revision and provided detailed answers to each of the reviews separately. R1, R2 and R4 did not participate in the discussion, and R5 stayed with the positive rating.
AC agrees with R5 that the provided analysis is insightful, and the effort put into organizing the research community around a single set of benchmarks and metrics is indeed valuable. However, given a simplistic nature of the proposed datasets and lack of other methodological contributions, the submission is not meeting the acceptance bar for ICLR. After discussion with PCs, the final recommendation is to reject.